# Consensus Optimization at Representation: Improving Personalized Federated Learning via Data-Centric Regularization

## Abstract

Federated learning is a large scale machine learning training paradigm where data is distributed across clients, and can be highly heterogeneous from one client to another. To ensure personalization in client models, and at the same time to ensure that the local models have enough commonality (i.e., prevent "client-drift"), it has been recently proposed to cast the federated learning problem as a consensus optimization problem, where local models are trained on local data, but are forced to be similar via a regularization term. In this paper we propose an improved federated learning algorithm, where we ensure consensus optimization at the *representation* part of each local client, and not on whole local models. This algorithm naturally takes into account that today's deep networks are often partitioned into a feature extraction part (representation) and a prediction part. Our algorithm ensures greater flexibility compared to previous works on exact shared representation in highly heterogeneous settings, as it has been seen that the representation part can differ substantially with data distribution. Our method is quite stable to noise, and can be made differentially private with strong privacy guarantee without much loss of accuracy. We provide a complete convergence analysis of our algorithm under general nonconvex loss functions, and validate its good performance experimentally in standard datasets.

## 1 Introduction

Federated learning (FL) has attracted much attention from the machine learning community recently due to rapid development of distributed intelligent devices and the demand of data privacy protection in large scale learning models. A typical FL framework is a machine learning training paradigm that includes a central server to aggregate the local information from participating clients to update a global model. The local data of each client should not be shared with other clients and should ideally be kept private up to certain degree also from the server Konečný et al. (2016); McMahan et al. (2017); Kairouz & McMahan (2021). With $M$ clients, a standard FL algorithm usually tries to solve the following optimization problem:

$$\min_{\omega} \frac{1}{M} \sum_{i=1}^{M} f_i(\omega) \tag{1}$$

where $\omega$ is a global model updated at the server, $f_i(\omega)$ is the local objective function at $i$-th client (the *empirical risk functions* at each of the client evaluated at their respective data samples). At each iteration, a local (stochastic) gradient or the entire local model is sent to the server for global model update.

However, in the context of FL, the data distribution across different clients are usually highly non-identical and heterogeneous. Thus in many practical applications, a single global model is not sufficient to satisfy the requirements of all the clients. To tackle this issue, many personalized FL methods have been proposed to allow each client to maintain a local model. A popular formulation of the problem is to use the concept of consensus optimization Smith et al. (2017); T Dinh et al. (2020); Li et al. (2021), that replaces the optimization problem of eq. (1) with the following:

$$\min_{\omega_0, \{\omega_i\}_{i=1}^{M}} \frac{1}{M} \sum_{i=1}^{M} f_i(\omega_i) + \frac{\lambda}{2} \|\omega_i - \omega_0\|^2 \tag{2}$$

where $\omega_0$ is the global model maintained at the server, $\omega_i$ is an unique local model at $i$-th client, and $\lambda$ is a hyper-parameter to balance the local training and forced consensus. The local models are not required

to be the exactly same but the regularization forces them to be close to each other, and the parameter $\lambda$ provides a flexibility to fit local data distribution.

Recent success of centralized multi-task learning is based on the realization that different tasks have shared common representation Bengio et al. (2013); Collins et al. (2021). Inspired by this observation, several studies have tried to exploit shared representation in personalized federated learning to achieve better local performance Arivazhagan et al. (2019); Collins et al. (2021); Pillutla et al. (2022). In this setting, at a high level, the local prediction model at each client is divided into two parts, including a representation part common to all clients. This motivates our first question:

*Q1: Can we force the consensus (cf. eq. 2) on the representation level, not the whole model level?*

Note that, a regularization at the representation part will include less number of variables, and therefore potentially is less expensive (e.g., in taking gradients) than a constraint on entire model. Indeed, in modern machine learning tasks, the model is usually a deep neural network consisting of a feature extractor and a prediction head. In the personalized FL works mentioned above, the deep neural network model is partitioned into a feature extractor $u$ and prediction head $v$. They consider the following optimization problem across different clients:

$$\min_{u,\{v_i\}_{i=1}^M} \frac{1}{M}\sum_{i=1}^M f_i(u,v_i) \tag{3}$$

where $u$ is a global feature extractor mapping inputs to a low dimensional space, $v_i$ is the local prediction head at $i$-th client. The server only maintains the global feature extractor $u$, not the whole model, and broadcasts it to all the clients at each communication round. The global extractor is trained via a method similar to FedAvg, a popular federated learning method Li et al. (2020b), and local prediction heads are trained only locally. Each client will generate the same representation for the same input. This method decouples the representation part and prediction part and obtains better performance on heterogeneous data Collins et al. (2021); Pillutla et al. (2022). However, as we will show in the next section, for different data distributions even the feature extractors can be different across clients. Although the prediction head has the largest difference between different clients, the differences in previous layers also exist Li et al. (2023). This motivates our second question:

*Q2: Can we further allow the feature extractor in one client to be different from others while still learning information on shared representations from other clients?*

Motivated by the two questions, in this work we propose a consensus optimization problem at the representation level as:

$$\min_{\{u_i\}_{i=0}^M,\{v_i\}_{i=1}^M} \frac{1}{M}\sum_{i=1}^M f_i(u_i,v_i) + \frac{\lambda}{2}\frac{1}{M}\sum_{i=1}^M H_i(u_i,u_0) \tag{4}$$

where $u_i$ and $v_i$ are the local feature extractor and local prediction head at $i$-th client, $i = 1,...,M$, respectively, $u_0$ is a global feature extractor maintained at the server. $H_i(u_i,u_0)$ is a regularization term to force the representation of the $i$th client $u_i$, which is defined on local dataset, and $u_0$ to be close. In this formulation the local feature extractors are no longer exactly same for each client, which provides more flexibility to fit highly heterogeneous data. The local models are almost trained locally except that their intermediate representations are forced to be close to each other. The local parameters are not covered by the global parameters in the training process, retaining more local knowledge. Fig. 1 displays an overview of our proposed framework. One point that we would like to stress: our regularization of the representation part is data-driven (compare with the regularization term in eq. 2).

For this formulation of the problem, we propose a new federated learning algorithm (detailed in Algorithm 4.1 and outlined in Fig. 1) based on distributed stochastic gradient descent. Note that, the server does not have the training data; the regularization term is defined based on the local data at $i$-th client. The server maintains $u_0$, which can be seen as a 'probe model' from server to detect the representations of local data.

As it is expensive to get access to the full gradient of $H_i(u_i,u_0)$, we leverage the batch of samples used to calculate the *stochastic gradient* of $f_i(u_i,v_i)$ to compute a stochastic gradient of $H_i(u_i,u_0)$. It will bring an additional stochastic noise of regularization term. To handle this issue, we further propose a partial variance (partial, because it is only applied to the stochastic gradient term related to regularization) reduction method to reduce the effect of the stochastic regularization. Note that, it has been previously observed that the

application of variance reduction methods in neural network training is not successful Defazio & Bottou (2019); Reddi et al. (2021); Li et al. (2023) - therefore we wish to retain the randomness of stochastic gradient of $f_i(u_i, v_i)$. As a result, we only apply the variance reduction technique on the stochastic regularization term. To avoid digression, this partial variance reduction part is delegated to Appendix C.

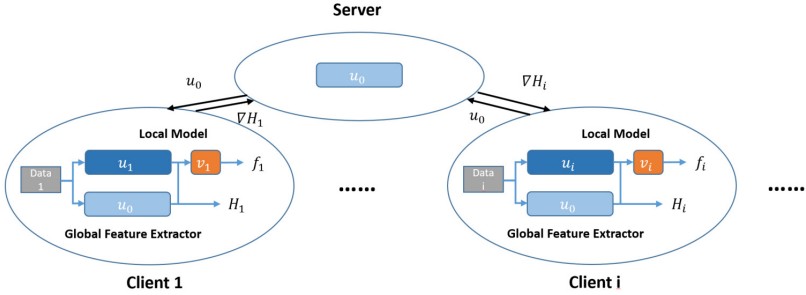

Figure 1: An overview of FedReCo.

Our contributions are summarized as follows.

- We propose a formulation of consensus optimization problem at the representation level to improve the flexibility of personalized federated learning (Sec. 3). The local models are trained locally except that their intermediate representations are forced to be close to each other by interacting with the server, retaining local knowledge to the maximum extent.

- We propose a stochastic gradient descent (SGD) based algorithm to solve this *representation consensus problem*, abbreviated as FedReCo (**Fed**erated **Re**presentation **Co**nsensus). Then we provide a theoretical convergence analysis of FedReCo for general non-convex functions (Sec. 4 and Sec. 5).

- Our algorithm is naturally private because clients share only their representation part with the server. Moreover, it is very noise resilient, and as a result can be easily adapted to a differential private variant to further protect data privacy, without loss of accuracy (see, Sec. 4.2 and Sec. 6.2.

- We conduct experiments on several benchmark datasets to illustrate the effectiveness of our proposed algorithms. Our algorithm can outperform the existing methods in highly heterogeneous settings (Sec. 6.1).

## 2 RELATED WORKS

**Personalized FL.** There are many strategies to achieve personalization in federated learning, including local fine-tuning Wang et al. (2019); Collins et al. (2022), meta-learning Chen et al. (2018); Jiang et al. (2019); Fallah et al. (2020), multi-task learning Smith et al. (2017), mixture of local and global model Hanzely & Richtárik (2020); Deng et al. (2020); Mansour et al. (2020), consensus based regularization T Dinh et al. (2020); Li et al. (2021). In all of these methods, the model is considered as a whole and fully personalized at each client.

**Consensus Based Regularization in FL.** Consensus based regularization has recently been applied in federated learning to force the local models to be close to each other. Notable works include FedProx Li et al. (2020a), that adds a proximal term to make the local model close to global model during the local training process, and Ditto Li et al. (2021), that has extended this idea to personalized federated learning. In addition, pFedMe T Dinh et al. (2020) proposes a bi-level problem based on similar proximal term and Moreau envelop as clients' loss function. In Li et al. (2019); Zhu & Ling (2022), $\ell_1$-norm based regularization has been proposed and proven to be robust to malicious attacks. Consensus optimization has also been studied in federated learning from a primal-dual view Zhang et al. (2021), including the alternating direction method of multipliers (ADMM) Zhou & Li (2021); Huang et al. (2019). The regularization term in these works is generally based on the difference between local model and global model in the whole model level.

**Shared Representation in FL.** The idea of partitioning a neural network into feature extractor and personalized prediction head has been applied to federated learning in Arivazhagan et al. (2019), which personalizes the last layer for different clients. The shared representation in linear regression problem and a convergence analysis has been given in Collins et al. (2021). Oh et al. (2022) only updates the feature

extractor with a randomly initialized prediction head, which is never updated in the training. Different from the shared feature extractor, the work of Liang et al. (2020) personalizes the first few layers and aggregates the last layer globally. Pillutla et al. (2022) considers a general framework of partial personalization in neural network training and establishes general convergence analysis for non-convex functions. Zhong et al. (2023) has extended the shared representation idea from different clients to different domains. Further, Shen et al. (2022) analyzes the differential privacy property for shared representation in federated learning.

On top of shared representation, Xu et al. (2023) and Zhang et al. (2023) add a regularization term in the local training of shared feature extractor. These two works are the ones most related to this work. Specifically, Xu et al. (2023) exploits the centroid of the representations within one class to regularize the local training, while Zhang et al. (2023) uses the difference between global and local *mutual information* as the regularization term. The primary differences of our work compared to these are: 1) The feature extractor in Xu et al. (2023) and Zhang et al. (2023) is still the same for every clients, although a regularization term is provided to constrain the update of feature extractor; 2) It is hard to provide privacy analysis on the regularization terms based on centroid of representations and mutual information. Being an SGD-type method our algorithm is on the other hand easily adapted to differential private versions; and 3) These prior works have not provided convergence analysis of their algorithm, while in this work we theoretically prove the convergence for our algorithms.

## 3 MOTIVATION AND PROBLEM STATEMENT

### 3.1 REPRESENTATION SIMILARITY ACROSS CLIENTS IN DIFFERENT LAYERS

In federated learning, the heterogeneous data distribution can lead the local models to different directions through multiple local SGD steps. The FedAvg-like algorithms suffer from this "client drift", which makes the local model far from global model within one communication round. In what follows, we will show the influence of client drift on the representations of different layers in one neural network model. We conduct an experiment on CIFAR10 dataset with a small 5-layer CNN model and ResNet18 He et al. (2016). There are 10 clients, each with 2 classes of data in the CIFAR10 dataset. We train the models via the FedAvg algorithm Li et al. (2020b). For each communication round, we perform two epochs of local SGD updates of the local model. After local iterations within one communication round $t$, we measure the similarity between the representations of local model $\omega_i^t$ and global model $\omega^t$ before aggregation. We use the centered kernel alignment (CKA) measurement Kornblith et al. (2019); Nguyen et al. (2021); Li et al. (2023) to quantify the similarity of representations.

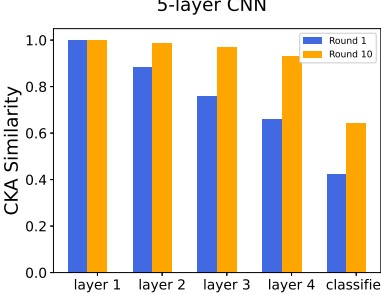
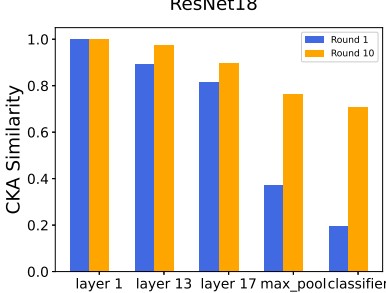

Figure 2: CKA Similarity for different layers.

Fig. 2 display the CKA similarity of representations after 1 round and 10 rounds of training, respectively. After only 1 round local training, the similarity decreases with deeper layers for both models. When training continues, the similarity between local model and global model increases for all the layers. After 10 rounds of training, the first four layers of 5-layer CNN model become close to global model, while the similarity of last (classifier) layer is still low. It shows that the FedAvg can learn a shared representation before the final classifier layer. However, even the previous layers are slightly dissimilar (similarity strictly less than 1). It is more obvious for larger ResNet18 model. Even after many rounds of training, the similarity decreases with deeper layers. The same phenomena has also been observed in Li et al. (2023) for a VGG model.

This observation motivates our work to consider a framework to allow different feature extractors in different clients while still learning the shared representations. The classifier layer or prediction head has the largest difference between local model and global model, thus we wish to train it completely locally. For the feature extractor, we train it locally, but with a regularization term to force it to learn representations from a global model.

## 3.2 REPRESENTATION CONSENSUS OPTIMIZATION PROBLEM

Let us know formally formulate our problem. Consider a federated learning system with $M$ clients, each client $i$ with $N$ samples $\{x_{ij} \in \mathbb{R}^{d_x}, y_{ij} \in \mathbb{R}^{d_y}\}_{j=1}^N, i=1,...,M$. For a designed neural network, the model is partitioned into a feature extractor $u$ and a prediction head $v$. For $i$-th client, it maintains its own local model $u_i$ and $v_i$. And the server maintains a global feature extractor $u_0$. If we are to mandate that the representations of local data to be the same for local feature extractor and global feature extractor, then the representation consensus optimization problem would be,

$$\min_{u_0, \{u_i, v_i\}_{i=1}^M} \sum_{i=1}^M f_i(u_i, v_i) \quad \text{s.t.} \ h_{ij}(u_i) = h_{ij}(u_0), \ i=1,2,...,M, \ j=1,2,...,N$$

where $f_i(u_i, v_i) = \frac{1}{N} \sum_{j=1}^N f_i(u_i, v_i | x_{ij}, y_{ij})$ is the empirical loss function with $N$ samples, and $h_{ij}(u) \triangleq h_{ij}(x_{ij}|u)$ is the mapping function $\mathbb{R}^{d_x} \to \mathbb{R}^p$ that maps input $x_{ij}$ to a intermediate representation with dimension $p$.

However we do not need the representations to be the exactly same for local feature extractor and global feature extractor. Thus we only put a $\ell_2$-norm regularization term to constrain the local training:

$$\min_{u_0, \{u_i, v_i\}_{i=1}^M} F(u_0, \{u_i, v_i\}_{i=1}^M) \triangleq \frac{1}{M} \sum_{i=1}^M f_i(u_i, v_i) + \frac{\lambda}{2} \frac{1}{M} \sum_{i=1}^M H_i(u_i, u_0) \tag{5}$$

where $H_i(u_i, u_0) = \frac{1}{N} \sum_{j=1}^N \|h_{ij}(u_i) - h_{ij}(u_0)\|^2$ is the regularization term to force the representations of all the local data samples to be close for local feature extractors and global feature extractor.

Since $f_i(u_i, v_i)$ and $H_i(u_i, u_0)$ are separable for each client $i$, we can also write the objective function as $F(u_0, \{u_i, v_i\}_{i=1}^M) = \frac{1}{M} \sum_{i=1}^M F_i(u_0, u_i, v_i)$ where $F_i(u_0, u_i, v_i) = f_i(u_i, v_i) + \frac{\lambda}{2} H_i(u_i, u_0)$.

The regularization term $H_i(u_i, u_0)$ is fully defined on the local dataset at $i$-th client. The server cannot know the local data samples and can only send the global feature extractor to the clients to "detect" local information.

## 4 FEDRECO ALGORITHM

### 4.1 ALGORITHM DESCRIPTION

Since both $f_i(u_i, v_i)$ and $H_i(u_i, u_0)$ are based on the local data samples, we can apply stochastic gradient descent (SGD) to solve the problem of eq. (5). We can exploit the same batch of data samples to calculate the stochastic gradients of $f_i(u_i, v_i)$ and $H_i(u_i, u_0)$ simultaneously, just passing the same batch of samples twice to model $\{u_i, v_i\}$ and feature extractor $u_0$, respectively. The global feature extractor only appears in the regularization term $H_i(u_i, u_0)$; thus we can just send the stochastic gradient of $H_i(u_i, u_0)$ with respect to $u_0$ to the server to update $u_0$ (this leads to a faster algorithm). To reduce the communication burden, we can perform multiple local SGD steps before transmitting the stochastic gradient. And due to the decoupling of feature extractor and prediction head, we can apply different learning rates and numbers of local steps to the two parts, respectively. In the following we use symbol $\tilde{\nabla}$ to represent stochastic gradient.

Specifically, our proposed FedReCo (Representation Consensus) algorithm is as follows: At each communication round $t$, the server broadcasts the $u_0$ to all the clients. Each client first updates the local prediction head $v_i$ via $K_v$ SGD local steps with learning rate $\eta_v$ as:

$$v_i^{t+1} = v_i^t - \eta_v \sum_{k=0}^{K_v - 1} \tilde{\nabla}_{v_i} f_i(v_i^{t,k}, u_i^t), \tag{6}$$

---

**Algorithm 1** FedReCo Algorithm

---

**Input**: Step size $\eta_u, \eta_v, \eta_0$, penalty parameter $\lambda$
**Initialize**: Initialize $u_0^0$ for server, initialize $u_i$ and $v_i$ for $i$-th client

1: **for** $t = 0, 1, ..., T-1$ **do**
2:    **Server**:
3:    Broadcast $u_0^t$ to all the clients
4:    Receive stochastic gradient $\tilde{\nabla}_{u_0} H_i(u_i^{t+1}, u_0^t)$ from all the clients
5:    Update $u_0$: $u_0^{t+1} = u_0^t - \eta_0 \frac{1}{M} \sum_{i=1}^{M} \tilde{\nabla}_{u_0} H_i(u_i^{t+1}, u_0^t)$
6:    **client** $i$:
7:    Receive $u_0^t$ from server, let $u_i^{t,0} = u_i^t$
8:    **for** $k = 0, 1, ..., K_v - 1$ **do**
9:       Randomly select one batch of samples, pass the samples to the model $\{u_i^t, v_i^{t,k}\}$ and calculate
         stochastic gradient $\tilde{\nabla}_{v_i} f_i(v_i^{t,k}, u_i^t)$
10:      Update $v_i^{t,k+1} = v_i^{t,k} - \eta_v \tilde{\nabla}_{v_i} f_i(v_i^{t,k}, u_i^t)$
11:    **end for**
12:    Let $v_i^{t+1} = v_i^{t,K_v}$
13:    **for** $k = 0, 1, ..., K_u - 1$ **do**
14:      Randomly select one batch of samples, pass the samples to model $\{u_i^{t,k}, v_i^{t+1}\}$ and calculate
         stochastic gradients $\tilde{\nabla}_{u_i} f_i(v_i^{t+1}, u_i^{t,k})$, pass the same batch of t samples to feature extractor $u_0^t$
         and calculate stochastic gradient $\tilde{\nabla}_{u_i} H_i(u_i^{t,k}, u_0^t)$
15:      Update $u_i^{t,k+1} = u_i^{t,k} - \eta_u \left( \tilde{\nabla}_{u_i} f_i(v_i^{t+1}, u_i^{t,k}) + \frac{\lambda}{2} \tilde{\nabla}_{u_i} H_i(u_i^{t,k}, u_0^t) \right)$
16:    **end for**
17:    Let $u_i^{t+1} = u_i^{t,K_u}$
18:    Randomly select one batch of samples and pass them to $u_i^{t+1}$ and $u_0^t$, calculate the stochastic
      gradient $\tilde{\nabla}_{u_0} H_i(u_i^{t+1}, u_0^t)$ and send it to the server
19: **end for**

---

where $v_i^{t,0} \equiv v_i^t$. Then the client fixes $v_i$ and updates local feature extractor $u_i$ via $K_u$ local steps with learning rate $\eta_u$ as

$$u_i^{t+1} = u_i^t - \eta_u \sum_{k=0}^{K_u-1} \left( \tilde{\nabla}_{u_i} f_i(v_i^{t+1}, u_i^{t,k}) + \frac{\lambda}{2} \tilde{\nabla}_{u_i} H_i(u_i^{t,k}, u_0^t) \right), \tag{7}$$

where again $u_i^{t,0} \equiv u_i^t$. After local training, the client calculates the stochastic gradient $\tilde{\nabla}_{u_0} H_i(u_i^{t+1}, u_0^t)$ and sends it to server. The server aggregates the stochastic gradients and updates $u_0$ with a server learning rate $\eta_0$ as

$$u_0^{t+1} = u_0^t - \eta_0 \frac{1}{M} \frac{\lambda}{2} \sum_{i=1}^{M} \tilde{\nabla}_{u_0} H_i(u_i^{t+1}, u_0^t). \tag{8}$$

The details of FedReCo algorithm are described in Algorithm 4.1.

## 4.2 Computation, Communication, and Privacy

For local training in FedReCo, each client needs to pass the same batch of samples to two models and calculate the stochastic gradients. The local training burden does not increase too much compared to FedAvg. Although the global feature extractor enlarges the demand of local computation and memory, the local computation power is not usually the bottleneck in the whole system. For the communication stage, each client needs to send a stochastic gradient $\tilde{\nabla}_{u_0} H_i(u_i^{t+1}, u_0^t)$ to the server, which only includes the gradients of model parameters in feature extractor, less than the whole model. It is like a sparsification method on the gradient, but with fixed selected dimensions. Therefore the size of the transmitted information is actually much lighter than FedAvg and prior consensus optimization works, improving the communication efficiency.

The server only receives the gradient of the norm of difference between local representations and global representations, which makes it harder to directly infer the local data, ensuring a more private setting.

Nevertheless, we can still easily adapt the FedReCo algorithm to differential private version by injecting the Gaussian noise to the stochastic gradient $\tilde{\nabla}_{u_0} H_i(u_i^{t+1}, u_0^t)$. Since the noise is added to the gradient of a penalty term on the difference between global representations and local representations, not to the gradient of local training function itself, the influence of the noise is pretty small. We can see from the experiments (see Sec. 6.2) that our algorithm can accommodate very high level privacy requirement of differential privacy. The privacy analysis is a straightforward extension of the standard Gaussian mechanism, thus omitted here.

As noted in the related works, FedPAC Xu et al. (2023) and FedCR Zhang et al. (2023) also add regularization terms in shared feature extractor. However, both methods need to send additional information to the server, enlarging the communication burden. Plus FedCR needs to estimate the distribution of local representations, which requires much more local computations (see Sec. 6.1). Finally it is not straightforward to add noise to the centroid or mutual information of the distributions, hence both works are not amenable to a privacy protection mechanism.

## 5 CONVERGENCE ANALYSIS

In this section we provide a theoretical convergence analysis of our proposed FedReCo algorithm. We first give the assumptions needed for theoretical analysis of our algorithm, and then provide a convergence result.

**Assumption 1** (Smoothness). We assume the smoothness of the loss function with respect to different parameters in the local model and global feature extractor.

- For each $i = 1, 2, ..., M$, the gradient $\nabla_{u_i} f_i(u_i, v_i)$ is $L_{f_u}$-Lipschitz with respect to $u_i$ and $L_{f_{uv}}$-Lipschitz with respect to $v_i$. Similarly, for each $i = 1, 2, ..., M$, the gradient $\nabla_{v_i} f_i(u_i, v_i)$ is $L_{f_v}$-Lipschitz with respect to $v_i$ and $L_{f_{vu}}$-Lipschitz with respect to $u_i$.

- For each $i = 1, 2, ..., M$, the gradient $\nabla_{u_i} H_i(u_i, u_0)$ is $L_{H_u}$-Lipschitz with respect to $u_i$ and $L_{H_{uu}}$-Lipschitz with respect to $u_0$. Similarly, the gradient $\nabla_{u_0} H_i(u_i, u_0)$ is $L_{H_u}$-Lipschitz with respect to $u_0$ and $L_{H_{uu}}$-Lipschitz with respect to $u_i$.

**Assumption 2** (Bounded variance of stochastic gradients). For each client $i = 1, 2, ..., M$, its stochastic gradient is unbiased and the variance of the stochastic gradient is upper-bounded by:

$$\mathbb{E}\left\|\tilde{\nabla}_{u_i} f_i(u_i, v_i) - \nabla_{u_i} f_i(u_i, v_i)\right\|^2 \leq \sigma_u^2, \quad \mathbb{E}\left\|\tilde{\nabla}_{v_i} f_i(u_i, v_i) - \nabla_{v_i} f_i(u_i, v_i)\right\|^2 \leq \sigma_v^2, \quad i = 1, ..., M$$

$$\mathbb{E}\left\|\tilde{\nabla}_{u_i} H_i(u_i, u_0) - \nabla_{u_i} H_i(u_i, u_0)\right\|^2 \leq \sigma_H^2, \quad \mathbb{E}\left\|\tilde{\nabla}_{u_0} H_i(u_i, u_0) - \nabla_{u_0} H_i(u_i, u_0)\right\|^2 \leq \sigma_H^2, \quad i = 1, ..., M.$$

The assumptions 1, 2 are standard in non-convex convergence analysis. Note that, we **do not** need a bound of heterogeneity since our method is full personalized and fits to arbitrary heterogeneous settings. We also **do not** need any bounded gradient assumption which is common in some literature.

In the following we let $U^t \triangleq [u_1^t, ..., u_M^t]$, and $V^t \triangleq [v_1^t, ..., v_M^t]$. For the measurements of convergence, we define

$$\Gamma_1^t = \|\nabla_{u_0} F(U^t, u_0^t)\|^2, \ \Gamma_2^t = \frac{1}{M} \sum_{i=1}^M \|\nabla_{u_i} F_i(u_i^t, v_i^t, u_0^t)\|^2, \ \Gamma_3^t = \frac{1}{M} \sum_{i=1}^M \|\nabla_{v_i} f_i(u_i^t, u_i^t)\|^2$$

If the three sequences converge to zero with $t$ in expectation, then we can obtain the convergence of $F$ in expectation to a stationary point. Throughout this paper we will denote by $F_{\min}$ the minimum of $F$ over its domain.

For the FedReCo algorithm described in Algorithm 4.1, we have the following convergence result.

**Theorem 1** (Convergence of FedReCo). *Suppose that Assumptions 1 and 2 hold. Let $L_1^2 = 2L_{f_u}^2 + \frac{\lambda^2}{2} L_{H_u}^2$, $\sigma_1^2 = 2\sigma_u^2 + \frac{\lambda^2 \sigma_H^2}{2}$. If learning rates satisfy $\eta_0 = \frac{\eta}{L_{H_u}}$, $\eta_u = \frac{\eta}{L_1 K_u}$, $\eta_v = \frac{\eta}{L_{f_v} K_v}$ and $\eta$ is chosen on the parameters $\lambda, L_{H_u}, L_1, L_{f_v}, L_{H_{uu}}, L_{f_{uv}}, \sigma_H, \sigma_u, \sigma_v$, then ignoring absolute constants, we have:*

$$\frac{1}{T} \sum_{t=0}^{T-1} \mathbb{E}\left(\frac{1}{4L_{H_u}} \Gamma_1^t + \frac{1}{16L_1} \Gamma_2^t + \frac{1}{8L_{f_v}} \Gamma_3^t\right) \lesssim \frac{\Sigma_1^{\frac{1}{2}}}{\sqrt{T}} + \frac{\Sigma_2^{\frac{1}{3}}}{T^{\frac{2}{3}}} + O\left(\frac{1}{T}\right) \tag{9}$$

*where*

$$\Sigma_1 = \frac{\lambda^3}{16M}\frac{\sigma_H^2}{L_{H_u}} + \frac{3}{2}\frac{\sigma_1^2}{L_1} + \frac{3}{2}\frac{\sigma_v^2}{L_{f_v}}, \quad \Sigma_2 = \frac{3}{20}\left(\lambda^2\frac{L_{H_{uu}}^2}{L_{H_u}L_1^2}\sigma_1^2 + \frac{L_{f_{uv}}^2}{L_1 L_{f_v}^2}\sigma_v^2\right).$$

*are positive constants depending on Lipschitz constants and stochastic variance.*

The proofs are deferred in Appendix B.

The left-hand side of (9) is a weighted sum of measurements $\Gamma_1, \Gamma_2, \Gamma_3$, that converges to zero with iterations. The decaying rate on the right-hand side is standard in non-convex SGD, and depends on the stochastic variances $\sigma_H, \sigma_u, \sigma_v$. In the $\Sigma_1$ at $O(1/\sqrt{T})$ term, $\sigma_H^2$ is divided by $M$, the number of clients. That's because $u_0$ is trained by aggregating the stochastic gradients from all the clients, while $u_i$ and $v_i$ are trained only locally.

The data-centric regularization term brings an additional error, which can be reduced by a partial variance reduction technique described in Appendix C. The $\sigma_H^2$ can be removed in the theoretical result of partial variance reduction, and we can further achieve $O(1/T)$ convergence rate by applying the variance reduction on both $f_i(u_i, v_i)$ function and regularization term $H_i(u_i, u_0)$. However, we found that in the practical training of neural network, the partial variance reduction brings almost no improvement. The performance of FedReCo is good even when the batch size is small, and sometimes slightly better than the variance-reduced version. This shows that FedReCo is robust to the additional error produced by the regularization term.

## 6 EXPERIMENTS

In this section we experimentally compare FedReCo with other recent personalized federated learning algorithms to show the effectiveness of our algorithm, and also show the privacy advantage of FedReCo.

### 6.1 PERFORMANCE ON BENCHMARK DATASETS

We perform the experiments on FashionMNIST/FMNIST and CIFAR10 datasets with a 5-layer CNN model, with two convolution layers and three fully connected layers. The first four layers are considered as the feature extractor and one last classifier layer as the prediction head trained totally locally. The compared methods include: FedAvg McMahan et al. (2017), FedAvg-FineTuning (FT) Collins et al. (2022), Ditto Li et al. (2021), FedRep Collins et al. (2021), FedBabu Oh et al. (2022), FedPAC Xu et al. (2023), FedCR Zhang et al. (2023). There are 50 clients in the network, each with 4 classes of data for FMNIST dataset and 2 classes of data for CIFAR10 dataset, to form a hetegenerous data distribution. The results are obtained after 500 rounds of communication, each with local SGD updates for 2 epochs of local samples, 1 epoch on training local prediction head, 1 epoch on training local feature extractor. More details of settings and hyper-parameters are provided in Appendix A.

For the relatively simple dataset FMNIST, FedAvg can already get an acceptable accuracy, and other algorithms obtain similar final accuracy. Note that in this case the FedAvg+fine-tuning is competitive to other methods, getting the highest accuracy. For the more complex CIFAR10 dataset and more heterogeneous setting, fine-tuning is still competitive to some personalization methods, with FedReCo outperforming all compared methods, showing the higher flexibility to more heterogeneous setting.

Table 1: Test Accuracy (%) on benchmark datasets; F: FMNIST, C: CIFAR10

|   | FEDAVG | FEDAVG-FT | DITTO | FEDREP | FEDBABU | FEDPAC | FEDCR | FEDRECO |
|---|--------|-----------|-------|--------|---------|--------|-------|---------|
| F | 85.38 | **93.85** | 92.92 | 93.04 | 92.85 | 92.62 | 92.71 | 93.09 |
| C | 56.17 | 89.05 | 90.55 | 89.12 | 85.69 | 88.71 | 89.21 | **91.07** |

To compare efficiency, Fig. 3 (a) displays the test accuracy of different algorithms with varying communication rounds, and Table 2 shows the running time of different algorithms when they achieve 85% accuracy on CIFAR10 dataset. The experiments are done in a single NVIDIA-A100-PCIE-40GB GPU, to simulate multiple clients. Note that, Table 2 only reflects the local computation cost, and does not include communication cost. It can be seen that Ditto can achieve the same accuracy with least time. FedReCo requires slightly more time in local computation as it needs to compute the representations twice for local model and global feature extractor, but gets higher final accuracy. Plus FedReCo communicates

less than Ditto which transmits the whole model. Compared to other methods using the partition of neural network, FedReCo is faster: FedPAC needs more rounds of iteration to achieve the same accuracy, and FedCR spends orders of magnitude more time on local computation.

Table 2: Running time to achieve 85% Accuracy on CIFAR10 dataset

| DITTO | FEDREP | FEDBABU | FEDPAC | FEDCR | FEDRECO |
|---|---|---|---|---|---|
| 52 MIN 41 S | 130 MIN 2 S | 317 MIN 51 S | 334 MIN 46 S | 1736 MIN 7 S | 61 MIN 56 S |

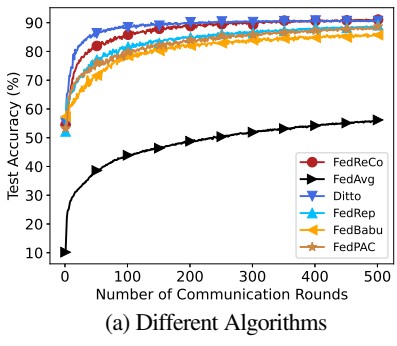
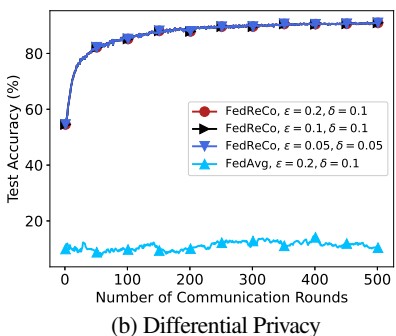

(a) Different Algorithms

(b) Differential Privacy

Figure 3: (a) Test accuracy on CIFAR10 dataset. (b) Test accuracy with differential privacy on CIFAR10 dataset.

## 6.2 ROBUSTNESS AND DIFFERENTIAL PRIVACY

We further explore the impact of differential privacy on our proposed algorithm. We use $(\epsilon,\delta)$-local differential privacy to reduce the risk of compromising local data and apply the standard Gaussian mechanism to add noise to the transmitted information Dwork & Roth (2014); Abadi et al. (2016). For FedReCo, we add Gaussian noise to the stochastic gradients of regularization term. To compare, we use FedAvg and add the Gaussian noise to the model difference within one round of local training, which is the information to be transmitted from a client to server for aggregation. More experiment details can be found in Appendix A.3.

Fig. 3 (b) shows the test accuracy with the number of communication rounds with Gaussian noise, and Table 3 displays the final accuracy of FedAvg, FedAvg-FT and FedReCo after 500 of communication rounds. We can see the FedReCo is almost not influenced by the added Gaussian noise, even when the $\epsilon$ and $\delta$ is pretty small, while FedAvg suffers a lot from the added noise. FedAvg with fine-tuning also suffers from the noise since the model trained by FedAvg cannot learn the local knowledge well with the added noise. This suggests a huge advantage to do optimization on the representation level, not at the model level, to be more robust to perturbations.

Table 3: Test Accuracy (%) with $(\epsilon,\delta)$-differential privacy

|  | FEDAVG | FEDAVG-FT | FEDRECO | |
|---|---|---|---|---|
|  | ($\epsilon{=}0.2,\delta{=}0.1$) | ($\epsilon{=}0.2,\delta{=}0.1$) | ($\epsilon{=}0.2,\delta{=}0.1$) | ($\epsilon{=}0.05,\delta{=}0.05$) |
| FMNIST | 8.43 | 69.00 | 93.14 | 93.15 |
| CIFAR10 | 10.42 | 64.25 | 90.95 | 90.93 |

## 7 CONCLUSIONS

We have proposed a federated learning algorithm, FedReCo, that enforces the representation part of local models to be similar in a data-driven manner. While being superior in accuracy and efficiency to many other methods, FedReCo is also noise-robust and can be made differentially private without degradation. FedReCo takes a step to study how layer sensitivity in neural networks can be fully exploited in federated learning, which hopefully will result in further interesting works. In fact, the framework of FedReCo can be easily extended to the partition of neural network at any layer, not limited to last classifier layer, and even to partitioning at different layers for different clients.

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
