## A  EXPERIMENT SETUP

### A.1  DATASETS

We use two benchmark datasets in the experiments, FashionMNIST (FMNIST) and CIFAR10, both consisting of 10 classes of data. The samples are divided into a training set with 70% data, and a testing set with 30% data. The data samples are distributed to 50 clients. To make the data distribution heterogeneous, we assign different classes of data to the clients. For FMNIST, each client has the data from 4 classes, and for CIFAR10, each client only has the data from 2 classes.

Table 4: Detailed information about datasets

| DATASET | ALL SAMPLES | TRAINING SET | TEST SET | SAMPLES PER CLIENT | CLASSES PER CLIENT |
|---------|-------------|--------------|----------|--------------------|--------------------|
| FMNIST  | 70000       | 49000        | 21000    | 1400               | 4                  |
| CIFAR10 | 60000       | 42000        | 18000    | 1200               | 2                  |

### A.2  MODEL AND HYPER-PARAMETERS

The model for both datasets is a 5-layer CNN model, consisting of two convolutional layers, each followed by a $2 \times 2$ max pooling layers, and two fully connected layers with 1024 neurons, and finally a softmax layer as classifier. The first four layers are considered as the feature extractor for FedRep, FedBabu, FedPAC, FedCR, FedReCo.

For the hyper-parameters in different algorithms, we set the same learning rates as 0.01, and batch size as 48. In the local training, the standard gradient clipping is used with a maximum norm 10. For FedReCo, we use the same learning rate for feature extractor and prediction head as 0.01, and 0.001 for the global feature extractor. The $\lambda$ in the regularization term is 0.01. For Ditto, the $\lambda$ is set as 0.01. For FedPAC, the $\lambda$ is set to 1. For FedCR, the $\beta$ is set as 0.001.

### A.3  DIFFERENTIAL PRIVACY

Here we describe how we add Gaussian noise to FedReCo and FedAvg algorithms for privacy. We aim to use local differential privacy to protect the information from client to the server, to reduce the risk that the server infers the local data in clients. Let us first, give a formal definition of $(\epsilon,\delta)$-local differential privacy.

**Definition 1.** Let $\mathcal{A}$ be a randomized algorithm that takes a client's private data as input. Let $\mathrm{im}(\mathcal{A})$ denote the image of $\mathcal{A}$. The algorithm $\mathcal{A}$ is said to provide $(\epsilon,\delta)$-local differential privacy if, for all pairs of clients' possible private data $x$ and $x'$ and all subsets $S$ of $\mathrm{im}(\mathcal{A})$:

$$\Pr[\mathcal{A}(x) \in S] \leq e^{\epsilon} \times \Pr[\mathcal{A}(x') \in S] + \delta$$

In the experiments we apply the standard Gaussian mechanism to add $\mathcal{N}(0,\sigma^2)$ noise to each element of the transmitted vector, where

$$\sigma = \sqrt{2\ln(1.25/\delta)}\frac{\Phi}{\epsilon}$$

and $\Phi$ is the sensitivity of the function to be added the noise. In the training of neural network, the sensitivity is the maximum norm of the gradients Abadi et al. (2016) and is given by gradient clipping. In the local training, the gradients are clipped with a maximum norm, which is used as sensitivity in differential privacy.

**FedReCo.** Each client needs to transmit stochastic gradient $\tilde{\nabla} H_i(u_i,u_0)$ to the server in one communication round. The clients add Gaussian noise to the stochastic gradient and send it to server. The server then aggregates the perturbed stochastic gradients, and also clips it within maximum norm to update the global feature extractor $u_0$. The update of $u_0$ is

$$u_0^{t+1} = u_0^t - \eta_0 \mathrm{Clip}\left[\frac{1}{M}\sum_{i=1}^{M}\left(\tilde{\nabla} H_i(u_i^{t+1},u_0^t) + g_i^t\right)\right]$$

where $g_i^t$ is the Gaussian noise, maximum norm (for clipping) is 10 for all the clients and server.

**FedAvg.** In FedAvg, at each communication round, one client receives the global model $\omega^t$ from the server and performs multiple local SGD steps to update the local model $\omega_i$. Then it sends the model difference $\Delta_i^t = \omega^t - \omega_i^{t+1}$ to the server. The model difference can be seen as a gradient to update the global model in (generalized) FedAvg. Now the client adds the Gaussian noise to the model difference and sends it to the server. Then the server aggregates the model difference and clips it to maximum norm to update the model. The update of global model is

$$\omega^{t+1} = \omega^t - \text{Clip}\left[\frac{1}{M}\sum_{i=1}^{M}(\Delta_i^t + g_i^t)\right]$$

where $g_i^t$ is the Gaussian noise, maximum norm is 10 for all the clients and server.

## B  PROOF OF FEDRECO CONVERGENCE

### B.1  PROOF OUTLINE OF THEOREM 1

In this section we give a proof of the Theorem 1. Some detailed proofs of technical lemmas can be found in the following sections.

The proof starts from the smoothness of the cost function $F(U^t, V^t, u_0^t)$. There are three groups of parameters to be updated, thus we break the cost function as three parts and use the smoothness bound respectively.

$$
\begin{aligned}
&F(U^{t+1}, V^{t+1}, u_0^{t+1}) - F(U^t, V^t, u_0^t) = \\
&\underbrace{F(U^{t+1}, V^{t+1}, u_0^{t+1}) - F(U^{t+1}, V^{t+1}, u_0^t)}_{D_1} + \underbrace{F(U^{t+1}, V^{t+1}, u_0^t) - F(U^t, V^{t+1}, u_0^t)}_{D_2} \\
&+ \underbrace{F(U^t, V^{t+1}, u_0^t) - F(U^t, V^t, u_0^t)}_{D_3}
\end{aligned}
\tag{10}
$$

In the following we first bound the $D_1, D_2, D_3$ respectively.

**Lemma 1.** *When $\eta_0 \le \frac{2}{\lambda L_{H_u}}$, the expectation of $D_1$ is bounded by*

$$\mathbb{E}D_1 \le -\frac{\eta_0}{2}\|\nabla_{u_0}F(U^{t+1}, u_0^t)\|^2 + \frac{\lambda^3 L_{H_u}\eta_0^2\sigma_H^2}{16M}$$

**Lemma 2.** *Let $L_1^2 = 2L_{f_u}^2 + \frac{\lambda^2}{2}L_{H_u}^2$, $\sigma_1^2 = 2\sigma_u^2 + \frac{\lambda^2\sigma_H^2}{2}$. When $\eta_u \le \frac{1}{8L_1 K_u}$, the expectation of $D_2$ satisfies*

$$\mathbb{E}D_2 \le \frac{1}{M}\sum_{i=1}^{M} -\frac{\eta_u K_u}{4}\|\nabla_{u_i}F(u_i^t, v_i^{t+1}, u_0^t)\|^2 + \frac{3}{2}\eta_u^2 K_u^2 L_1\sigma_1^2$$

**Lemma 3.** *When $\eta_v \le \frac{1}{8L_{f_v}K_v}$, the expectation of $D_3$ satisfies*

$$\mathbb{E}D_3 \le \frac{1}{M}\sum_{i=1}^{M} -\frac{\eta_v K_v}{4}\|\nabla_{v_i}f_i(u_i^t, v_i^t)\|^2 + \frac{3}{2}\eta_v^2 K_v^2 L_{f_v}\sigma_v^2$$

With the three lemmas, we can write

$$
\begin{aligned}
&\frac{\eta_0}{2}\mathbb{E}\|\nabla_{u_0}F(U^{t+1}, u_0^t)\|^2 + \frac{1}{M}\sum_{i=1}^{M}\frac{\eta_u K_u}{4}\mathbb{E}\|\nabla_{u_i}F_i(u_i^t, v_i^{t+1}, u_0^t)\|^2 + \frac{1}{M}\sum_{i=1}^{M}\frac{\eta_v K_v}{4}\mathbb{E}\|\nabla_{v_i}f_i(u_i^t, v_i^t)\|^2 \\
&\le \mathbb{E}\big[F(u_i^t, v_i^t, u_0^t) - F(u_i^{t+1}, v_i^{t+1}, u_0^{t+1})\big] + \frac{\lambda^3 L_{H_u}\eta_0^2\sigma_H^2}{16M} + \frac{3}{2}\eta_u^2 K_u^2 L_1\sigma_1^2 + \frac{3}{2}\eta_v^2 K_v^2 L_{f_v}\sigma_v^2
\end{aligned}
\tag{11}
$$

Note that the metric we use to measure the convergence is $\Gamma_1^t = \|\nabla_{u_0}F(U^t, u_0^t)\|^2$ and $\Gamma_2^t = \frac{1}{M}\sum_{i=1}^{M}\|\nabla_{u_i}F_i(u_i^t, v_i^t, u_0^t)\|^2$. Now the left-hand side (LHS) of (11) includes $\|\nabla_{u_0}F(U^{t+1}, u_0^t)\|^2$ and $\frac{1}{M}\sum_{i=1}^{M}\|\nabla_{u_i}F_i(u_i^t, v_i^{t+1}, u_0^t)\|^2$, which are different from our measurements. Thus we prove the following lemmas to close the gap.

**Lemma 4.** *When $\eta_u \leq \frac{1}{8L_1 K_u}$, the expectation of $\|\nabla_{u_0} F(U^t, u_0^t)\|^2$ is bounded by*

$$\mathbb{E}\|\nabla_{u_0} F(U^t, u_0^t)\|^2$$
$$\leq \frac{6}{5}\lambda^2 L_{H_{uu}}^2 \eta_u^2 K_u^2 \frac{1}{M}\sum_{i=1}^{M}\|\nabla_{u_i} F_i(u_i^t, v_i^{t+1}, u_0^t)\|^2 + \frac{3}{5}\lambda^2 L_{H_{uu}}^2 \eta_u^2 K_u^2 \sigma_1^2 + 2\|\nabla_{u_0} F(U^{t+1}, u_0^t)\|^2$$

**Lemma 5.** *When $\eta_v \leq \frac{1}{8L_{f_v K_v}}$, the expectation of $\|\nabla_{u_i} F_i(u_i^t, v_i^t, u_0^t)\|^2$ is bounded by*

$$\mathbb{E}\|\nabla_{u_i} F_i(u_i^t, v_i^t, u_0^t)\|^2$$
$$\leq \frac{24}{5}L_{f_{uv}}^2 K_v^2 \eta_v^2 \|\nabla_{v_i} f_i(u_i^t, v_i^t)\|^2 + \frac{12}{5}L_{f_{uv}}^2 \eta_v^2 K_v^2 \sigma_v^2 + 2\|\nabla_{u_i} F_i(u_i^t, v_i^{t+1}, u_0^t)\|^2$$

Using the two lemmas, we can get

$$\frac{\eta_0}{4}\|\nabla_{u_0} F(U^t, u_0^t)\|^2 + \frac{\eta_u K_u}{16}\frac{1}{M}\sum_{i=1}^{M}\|F_i(u_i^t, v_i^t, u_0^t)\|^2 + \frac{\eta_v K_v}{8}\frac{1}{M}\sum_{i=1}^{M}\|\nabla_{v_i} f_i(u_i^t, v_i^t)\|^2$$

$$\leq \frac{\eta_0}{2}\|\nabla_{u_0} F(U^{t+1}, u_0^t)\|^2 + \left(\frac{3}{10}\lambda^2 \eta_0 \eta_u^2 L_{H_{uu}}^2 K_u^2 + \frac{1}{8}\eta_u K_u\right)\frac{1}{M}\sum_{i=1}^{M}\|\nabla_{u_i} F_i(u_i^t, v_i^{t+1}, u_0^t)\|^2$$

$$+ \left(\frac{3}{10}L_{f_{uv}}^2 \eta_u K_u \eta_v^2 K_v^2 + \frac{1}{8}\eta_v K_v\right)\frac{1}{M}\sum_{i=1}^{M}\|\nabla_{v_i} f_i(u_i^t, v_i^t)\|^2$$

$$+ \frac{3}{20}\lambda^2 L_{H_{uu}}^2 \eta_0 \eta_u^2 K_u^2 \sigma_1^2 + \frac{3}{20}L_{f_{uv}}^2 \eta_u K_u \eta_v^2 K_v^2 \sigma_v^2$$

When $\lambda^2 \eta_0 \eta_u K_u L_{H_{uu}}^2 \leq \frac{5}{12}$ and $\eta_u K_u \eta_v K_v L_{f_{uv}}^2 \leq \frac{5}{12}$, after a calculation, we have

$$\frac{\eta_0}{4}\|\nabla_{u_0} F(U^t, u_0^t)\|^2 + \frac{\eta_u K_u}{16}\frac{1}{M}\sum_{i=1}^{M}\|F_i(u_i^t, v_i^t, u_0^t)\|^2 + \frac{\eta_v K_v}{8}\frac{1}{M}\sum_{i=1}^{M}\|\nabla_{v_i} f_i(u_i^t, v_i^t)\|^2$$

$$\leq \frac{\eta_0}{2}\mathbb{E}\|\nabla_{u_0} F(U^{t+1}, u_0^t)\|^2 + \frac{1}{M}\sum_{i=1}^{M}\frac{\eta_u K_u}{4}\mathbb{E}\|\nabla_{u_i} F_i(u_i^t, v_i^{t+1}, u_0^t)\|^2$$

$$+ \frac{1}{M}\sum_{i=1}^{M}\frac{\eta_v K_v}{4}\mathbb{E}\|\nabla_{v_i} f_i(u_i^t, v_i^t)\|^2 + \frac{3}{20}\lambda^2 L_{H_{uu}}^2 \eta_0 \eta_u^2 K_u^2 \sigma_1^2 + \frac{3}{20}L_{f_{uv}}^2 \eta_u K_u \eta_v^2 K_v^2 \sigma_v^2$$

$$\leq \mathbb{E}\left[F(u_i^t, v_i^t, u_0^t) - F(u_i^{t+1}, v_i^{t+1}, u_0^{t+1})\right] + \frac{\lambda^3 L_{H_u} \eta_0^2 \sigma_H^2}{16M}$$

$$+ \left(\frac{3}{2}\eta_u^2 K_u^2 L_1 + \frac{3}{20}\lambda^2 L_{H_{uu}}^2 \eta_0 \eta_u^2 K_u^2\right)\sigma_1^2 + \left(\frac{3}{2}\eta_v^2 K_v^2 L_{f_v} + \frac{3}{20}L_{f_{uv}}^2 \eta_u K_u \eta_v^2 K_v^2\right)\sigma_v^2 \qquad (12)$$

where the second inequality is from (11). Now the LHS of (12) includes $\Gamma_1^t, \Gamma_2^t, \Gamma_3^t$.

Let $\eta_0 = \frac{\eta}{L_{H_u}}$, $\eta_u = \frac{\eta}{L_1 K_u}$, $\eta_v = \frac{\eta}{L_{f_v} K_v}$. Then we can obtain

$$\frac{1}{4L_{H_u}}\mathbb{E}\|\nabla_{u_0} F(U^t, u_0^t)\|^2 + \frac{1}{16L_1}\frac{1}{M}\sum_{i=1}^{M}\|\nabla_{u_i} F_i(u_i^t, v_i^t, u_0^t)\|^2 + \frac{1}{8L_{f_v}}\frac{1}{M}\sum_{i=1}^{M}\|\nabla_{v_i} f_i(u_i^t, u_i^t)\|^2$$

$$\leq \frac{\mathbb{E}\left[F(u_i^t, v_i^t, u_0^t) - F(u_i^{t+1}, v_i^{t+1}, u_0^{t+1})\right]}{\eta} + \eta\left(\frac{\lambda^3}{16M}\frac{\sigma_H^2}{L_{H_u}} + \frac{3}{2}\frac{\sigma_1^2}{L_1} + \frac{3}{2}\frac{\sigma_v^2}{L_{f_v}}\right)$$

$$+ \frac{3\eta^2}{20}\left(\lambda^2 \frac{L_{H_{uu}}^2}{L_{H_u} L_1^2}\sigma_1^2 + \frac{L_{f_{uv}}^2}{L_1 L_{f_v}^2}\sigma_v^2\right)$$

Define $\Sigma_1 = \frac{\lambda^3}{16M}\frac{\sigma_H^2}{L_{H_u}} + \frac{3}{2}\frac{\sigma_1^2}{L_1} + \frac{3}{2}\frac{\sigma_v^2}{L_{f_v}}$ and $\Sigma_2 = \frac{3}{20}\left(\lambda^2\frac{L_{H_{uu}}^2}{L_{H_u}L_1^2}\sigma_1^2 + \frac{L_{f_{uv}}^2}{L_1L_{f_v}^2}\sigma_v^2\right)$. Applying telescopic cancellation through $t=0$ to $t=T-1$, we have

$$\frac{1}{T}\sum_{t=0}^{T-1}\left(\frac{1}{4L_{H_u}}\Gamma_1^t + \frac{1}{16L_1}\Gamma_2^t + \frac{1}{8L_{f_v}}\Gamma_3^t\right)$$

$$\leq \frac{\left[F(u_i^0,v_i^0,u_0^0) - F(u_i^T,v_i^T,u_0^T)\right]}{\eta T} + \eta\Sigma_1 + \eta^2\Sigma_2$$

$$\leq \frac{F(u_i^0,v_i^0,u_0^0) - F_{\min}}{\eta T} + \eta\Sigma_1 + \eta^2\Sigma_2$$

We need to make $\eta$ satisfy the conditions in all the above proofs, thus

$$\eta \leq \min\left\{\frac{2}{\lambda},\quad \frac{1}{8},\quad \sqrt{\frac{5L_{H_u}L_1}{12\lambda^2 L_{H_{uu}}^2}},\quad \sqrt{\frac{5L_{f_v}L_1}{12L_{f_{uv}}^2}}\right\}$$

Define $\Delta F = F(u_i^0,v_i^0,u_0^0) - F_{\min}$. Let

$$\eta = \frac{1}{\frac{\lambda}{2} + 8 + \sqrt{\frac{12\lambda^2 L_{H_{uu}}^2}{5L_{H_u}L_1}} + \sqrt{\frac{12L_{f_{uv}}^2}{5L_{f_v}L_1}} + \sqrt{\Sigma_1 T} + \Sigma_2^{\frac{1}{3}}T^{\frac{1}{3}}}$$

Then we can get

$$\frac{1}{T}\sum_{t=0}^{T-1}\mathbb{E}\left(\frac{1}{4L_{H_u}}\Gamma_1^t + \frac{1}{16L_1}\Gamma_2^t + \frac{1}{8L_{f_v}}\Gamma_3^t\right)$$

$$\leq \frac{(\Delta F + 1)\sqrt{\Sigma_1}}{\sqrt{T}} + \frac{(\Delta F + 1)\Sigma_2^{\frac{1}{3}}}{T^{\frac{2}{3}}} + \frac{\Delta F}{T}\left(\frac{\lambda}{2} + 8 + \sqrt{\frac{12\lambda^2 L_{H_{uu}}^2}{5L_{H_u}L_1}} + \sqrt{\frac{12L_{f_{uv}}^2}{5L_{f_v}L_1}}\right)$$

Ignoring absolute constants, we have

$$\frac{1}{T}\sum_{t=0}^{T-1}\mathbb{E}\left(\frac{1}{4L_{H_u}}\Gamma_1^t + \frac{1}{16L_1}\Gamma_2^t + \frac{1}{8L_{f_v}}\Gamma_3^t\right) \lesssim \frac{\Sigma_1^{\frac{1}{2}}}{\sqrt{T}} + \frac{\Sigma_2^{\frac{1}{3}}}{T^{\frac{1}{3}}} + O\left(\frac{1}{T}\right) \tag{13}$$

### B.2 PROOF OF LEMMA 1

*Proof.* The expectation of $D_1$ is

$$\mathbb{E}D_1 = \mathbb{E}F(U^{t+1},V^{t+1},u_0^{t+1}) - F(U^{t+1},V^{t+1},u_0^t)$$

$$= \mathbb{E}\frac{\lambda}{2M}\sum_{i=1}^{M}H_i(u_i^{t+1},u_0^{t+1}) - H_i(u_i^{t+1},u_0^t)$$

$$\leq \mathbb{E}\frac{\lambda}{2M}\sum_{i=1}^{M}\left(\langle\nabla_{u_0}H_i(u_i^{t+1},u_0^t),u_0^{t+1}-u_0^t\rangle + \mathbb{E}\frac{L_{H_u}}{2}\|u_0^{t+1}-u_0^t\|^2\right)$$

$$= -\mathbb{E}\eta_0\left\langle\sum_{i=1}^{M}\frac{\lambda}{2M}\nabla_{u_0}H_i(u_i^{t+1},u_0^t),\sum_{i=1}^{M}\frac{\lambda}{2M}\tilde{\nabla}_{u_0}H_i(u_i^{t+1},u_0^t)\right\rangle + \frac{\lambda L_{H_u}\eta_0^2}{4}\mathbb{E}\left\|\frac{\lambda}{2M}\sum_{i=1}^{M}\tilde{\nabla}_{u_0}H_i(u_i^{t+1},u_0^t)\right\|^2$$

$$\leq -\eta_0\left\|\sum_{i=1}^{M}\frac{\lambda}{2M}\nabla_{u_0}H_i(u_i^{t+1},u_0^t)\right\|^2 + \frac{\lambda L_{H_u}\eta_0^2}{4}\left\|\frac{\lambda}{2M}\sum_{i=1}^{M}\tilde{\nabla}_{u_0}H_i(u_i^{t+1},u_0^t)\right\|^2$$

where the inequality is from the smoothness of function $H_i$ with respect to $u_0$, the last equality is from the unbiasedness of stochastic gradient.

Then for the second term, we have

$$
\begin{aligned}
&\mathbb{E}\left\|\frac{\lambda}{2M}\sum_{i=1}^{M}\tilde{\nabla}_{u_0}H_i(u_i^{t+1},u_0^t)\right\|^2\\
=&\mathbb{E}\left\|\frac{\lambda}{2M}\sum_{i=1}^{M}\tilde{\nabla}_{u_0}H_i(u_i^{t+1},u_0^t)-\frac{\lambda}{2M}\sum_{i=1}^{M}\nabla_{u_0}H_i(u_i^{t+1},u_0^t)+\frac{\lambda}{2M}\sum_{i=1}^{M}\nabla_{u_0}H_i(u_i^{t+1},u_0^t)\right\|^2\\
=&\mathbb{E}\left\|\frac{\lambda}{2M}\sum_{i=1}^{M}\tilde{\nabla}_{u_0}H_i(u_i^{t+1},u_0^t)-\frac{\lambda}{2M}\sum_{i=1}^{M}\nabla_{u_0}H_i(u_i^{t+1},u_0^t)\right\|^2+\left\|\frac{\lambda}{2M}\sum_{i=1}^{M}\nabla_{u_0}H_i(u_i^{t+1},u_0^t)\right\|^2\\
=&\mathbb{E}\frac{\lambda^2}{4M^2}\sum_{i=1}^{M}\left\|\left(\tilde{\nabla}_{u_0}H_i(u_i^{t+1},u_0^t)-\nabla_{u_0}H_i(u_i^{t+1},u_0^t)\right)\right\|^2+\left\|\frac{\lambda}{2M}\sum_{i=1}^{M}\nabla_{u_0}H_i(u_i^{t+1},u_0^t)\right\|^2\\
\leq&\frac{\lambda^2\sigma_H^2}{4M}+\left\|\frac{\lambda}{2M}\sum_{i=1}^{M}\nabla_{u_0}H_i(u_i^{t+1},u_0^t)\right\|^2
\end{aligned}
$$

where the second and third equalities are from the unbiasedness of stochastic gradients, the inequality is from the Assumption 2.

Thus the $\mathbb{E}D_1$ can be bounded by

$$
\mathbb{E}D_1\leq-\eta_0\left(1-\frac{\lambda L_{H_u}\eta_0}{4}\right)\left\|\sum_{i=1}^{M}\frac{\lambda}{2M}\nabla_{u_0}H_i(u_i^{t+1},u_0^t)\right\|^2+\frac{\lambda^3 L_{H_u}\eta_0^2\sigma_H^2}{16M}
$$

When the learning rate $\eta_0$ satisfies

$$
\eta_0\leq\frac{2}{\lambda L_{H_u}}, \tag{14}
$$

we can obtain

$$
\begin{aligned}
\mathbb{E}D_1&\leq-\frac{\eta_0}{2}\left\|\sum_{i=1}^{M}\frac{\lambda}{2M}\nabla_{u_0}H_i(u_i^{t+1},u_0^t)\right\|^2+\frac{\lambda^3 L_{H_u}\eta_0^2\sigma_H^2}{16M}\\
&=-\frac{\eta_0}{2}\left\|\nabla_{u_0}F(U^{t+1},u_0^t)\right\|^2+\frac{\lambda^3 L_{H_u}\eta_0^2\sigma_H^2}{16M}
\end{aligned}
$$

$\square$

### B.3 PROOF OF LEMMA 2

*Proof.* The expectation of $D_2$ is

$$
\begin{aligned}
\mathbb{E}D_2 &= \mathbb{E}F(U^{t+1},V^{t+1},u_0^t) - F(U^t,V^{t+1},u_0^t) \\
&= \mathbb{E}\frac{1}{M}\sum_{i=1}^{M} F_i(u_i^{t+1},v_i^{t+1},u_0^t) - F_i(u_i^t,v_i^{t+1},u_0^t) \\
&= \mathbb{E}\frac{1}{M}\sum_{i=1}^{M} f_i(u_i^{t+1},v_i^{t+1}) - f_i(u_i^t,v_i^{t+1}) + \frac{\lambda}{2M}\sum_{i=1}^{M} H_i(u_i^{t+1},u_0^t) - H_i(u_i^t,u_0^t) \\
&\leq \mathbb{E}\frac{1}{M}\sum_{i=1}^{M}\langle\nabla_{u_i}f_i(u_i^t,v_i^t),u_i^{t+1}-u_i^t\rangle + \frac{L_{f_u}}{2}\|u_i^{t+1}-u_i^t\|^2 \\
&\quad + \frac{\lambda}{2M}\sum_{i=1}^{M}\langle\nabla_{u_i}H_i(u_i^t,u_0^t),u_i^{t+1}-u_i^t\rangle + \frac{L_{H_u}}{2}\|u_i^{t+1}-u_i^t\|^2 \\
&= \mathbb{E}\frac{1}{M}\sum_{i=1}^{M}\left\langle\nabla_{u_i}f_i(u_i^t,v_i^t)+\frac{\lambda}{2}\nabla_{u_i}H_i(u_i^t,u_0^t),u_i^{t+1}-u_i^t\right\rangle + \left(\frac{L_{f_u}}{2}+\frac{\lambda L_{H_u}}{4}\right)\|u_i^{t+1}-u_i^t\|^2 \\
&= \mathbb{E}\frac{1}{M}\sum_{i=1}^{M}\underbrace{\langle\nabla_{u_i}F_i(u_i^t,v_i^{t+1},u_0^t),u_i^{t+1}-u_i^t\rangle}_{G_i} + \left(\frac{L_{f_u}}{2}+\frac{\lambda L_{H_u}}{4}\right)\|u_i^{t+1}-u_i^t\|^2 \quad (15)
\end{aligned}
$$

where the inequality is from the smoothness of function $f_i$ and $H_i$.

For the first term of RHS, we have

$$
\begin{aligned}
\mathbb{E}G_i &= -\mathbb{E}\eta_u\left\langle\nabla_{u_i}F_i(u_i^t,v_i^{t+1},u_0^t), \sum_{k=0}^{K_u-1}\tilde{\nabla}_{u_i}F_i(u_i^{t,k},v_i^{t+1},u_0^t)\right\rangle \\
&= -\eta_u K_u\mathbb{E}\|\nabla_{u_i}F_i(u_i^t,v_i^{t+1},u_0^t)\|^2 \\
&\quad + \sum_{k=0}^{K_u-1}\mathbb{E}\eta_u\left\langle\nabla_{u_i}F_i(u_i^t,v_i^{t+1},u_0^t),\nabla_{u_i}F_i(u_i^{t,k},v_i^{t+1},u_0^t)-\nabla_{u_i}F_i(u_i^t,v_i^{t+1},u_0^t)\right\rangle \\
&\leq -\eta_u K_u\mathbb{E}\|\nabla_{u_i}F_i(u_i^t,v_i^{t+1},u_0^t)\|^2 \\
&\quad + \sum_{k=0}^{K_u-1}\frac{\eta_u}{2}\|\nabla_{u_i}F_i(u_i^t,v_i^{t+1},u_0^t)\|^2 + \frac{\eta_u}{2}\|\nabla_{u_i}F_i(u_i^{t,k},v_i^{t+1},u_0^t)-\nabla_{u_i}F_i(u_i^t,v_i^{t+1},u_0^t)\|^2 \\
&= -\frac{\eta_u K_u}{2}\|\nabla_{u_i}F_i(u_i^t,v_i^{t+1},u_0^t)\|^2 + \frac{\eta_u}{2}\sum_{k=0}^{K_u-1}\|\nabla_{u_i}F_i(u_i^{t,k},v_i^{t+1},u_0^t)-\nabla_{u_i}F_i(u_i^t,v_i^{t+1},u_0^t)\|^2 \quad (16)
\end{aligned}
$$

where the inequality is from the fact that $<x,y>\leq\frac{\|x\|^2}{2}+\frac{\|y\|^2}{2}$.

For the second term in (16), we can obtain

$$
\begin{aligned}
&\mathbb{E}\|\nabla_{u_i}F_i(u_i^{t,k},v_i^{t+1},u_0^t)-\nabla_{u_i}F_i(u_i^t,v_i^{t+1},u_0^t)\|^2 \\
&= \mathbb{E}\|\nabla_{u_i}f_i(u_i^{t,k},v_i^{t+1})-\nabla_{u_i}f_i(u_i^t,v_i^{t+1},u_0^t)+\frac{\lambda}{2}\nabla_{u_i}H_i(u_i^{t,k},u_0^t)-\frac{\lambda}{2}\nabla_{u_i}H_i(u_i^t,u_0^t)\|^2 \\
&\leq 2\mathbb{E}\|\nabla_{u_i}f_i(u_i^{t,k},v_i^{t+1})-\nabla_{u_i}f_i(u_i^t,v_i^{t+1},u_0^t)\|^2 + \frac{\lambda^2}{2}\mathbb{E}\|H_i(u_i^{t,k},u_0^t)-\nabla_{u_i}H_i(u_i^t,u_0^t)\|^2 \\
&\leq 2L_{f_u}^2\|u_i^{t,k}-u_i^t\|^2 + \frac{\lambda^2 L_{H_u}^2}{2}\|u_i^{t,k}-u_i^t\|^2 \\
&= \left(2L_{f_u}^2 + \frac{\lambda^2 L_{H_u}^2}{2}\right)\|u_i^{t,k}-u_i^t\|^2
\end{aligned}
$$

where the first inequality is from the fact that $\|x+y\|^2 \leq 2\|x\|^2 + 2\|y\|^2$, the second inequality is from the Lipschitz property of gradients.

We then plug the above inequality back to (16), and get

$$\mathbb{E}G_i \leq -\frac{\eta_u K_u}{2}\|\nabla_{u_i}F(u_i^t, v_i^{t+1}, u_0^t)\|^2 + \frac{\eta_u}{2}\sum_{k=1}^{K_u-1}\left(2L_{f_u}^2 + \frac{\lambda^2 L_{H_u}^2}{2}\right)\|u_i^{t,k} - u_i^t\|^2 \tag{17}$$

For the second term in RHS of (15), we can get

$$\mathbb{E}\|u_i^{t+1} - u_i^t\|^2 = \mathbb{E}\eta_u^2\|\sum_{k=0}^{K_u-1}\tilde{\nabla}_{u_i}F_i(u_i^{t,k}, v_i^{t+1}, u_0^t)\|^2$$

$$\leq \mathbb{E}K_u\eta_u^2\sum_{k=0}^{K_u-1}\|\tilde{\nabla}_{u_i}F_i(u_i^{t,k}, v_i^{t+1}, u_0^t)\|^2$$

$$= K_u\eta_u^2\sum_{k=0}^{K_u-1}\|\tilde{\nabla}_{u_i}F_i(u_i^{t,k}, v_i^{t+1}, u_0^t) - \nabla_{u_i}F_i(u_i^{t,k}, v_i^{t+1}, u_0^t)\|^2 + K_u\eta_u^2\sum_{k=0}^{K_u-1}\|\nabla_{u_i}F_i(u_i^{t,k}, v_i^{t+1}, u_0^t)\|^2 \tag{18}$$

where the equality is from the unbiasedness of stochastic gradient.

For the first term of RHS of (18), we have

$$\mathbb{E}\|\tilde{\nabla}_{u_i}F_i(u_i^{t,k}, v_i^{t+1}, u_0^t)\|^2$$

$$= \mathbb{E}\|\tilde{\nabla}f_i(u_i^{t,k}, v_i^{t+1}) - \nabla f_i(u_i^{t,k}, v_i^{t+1}) + \frac{\lambda}{2}\tilde{\nabla}H_i(u_i^{t,k}, v_i^{t+1}, u_0^t) - \frac{\lambda}{2}\nabla H_i(u_i^{t,k}, v_i^{t+1}, u_0^t)\|^2$$

$$\leq 2\mathbb{E}\|\tilde{\nabla}f_i(u_i^{t,k}, v_i^{t+1}) - \nabla f_i(u_i^{t,k}, v_i^{t+1})\|^2 + \frac{\lambda^2}{2}\|\tilde{\nabla}H_i(u_i^{t,k}, v_i^{t+1}, u_0^t) - \frac{\lambda}{2}\nabla H_i(u_i^{t,k}, v_i^{t+1}, u_0^t)\|^2$$

$$\leq 2\sigma_u^2 + \frac{\lambda^2}{2}\sigma_H^2 \tag{19}$$

where the first inequality is from $\|x+y\|^2 \leq 2\|x\|^2 + 2\|y\|^2$, the second inequality is from the Assumption 2.

For the second term of RHS of (18), we have

$$\|\nabla_{u_i}F_i(u_i^{t,k}, v_i^{t+1}, u_0^t)\|^2$$

$$\leq 2\|\nabla_{u_i}F_i(u_i^{t,k}, v_i^{t+1}, u_0^t) - \nabla_{u_i}F_i(u_i^t, v_i^{t+1}, u_0^t)\|^2 + 2\|\nabla_{u_i}F_i(u_i^t, v_i^{t+1}, u_0^t)\|^2$$

$$= 2\left\|\nabla_{u_i}f_i(u_i^{t,k}, v_i^{t+1}) - \nabla_{u_i}f_i(u_i^t, v_i^{t+1}) + \frac{\lambda}{2}\nabla_{u_i}H_i(u_i^{t,k}, u_0^t) - \frac{\lambda}{2}\nabla_{u_i}H_i(u_i^t, u_0^t)\right\|^2 + 2\|\nabla_{u_i}F_i(u_i^t, v_i^{t+1}, u_0^t)\|^2$$

$$\leq 2\left(2L_{f_u}^2 + \frac{\lambda^2}{2}L_{H_u}^2\right)\|u_i^{t,k} - u_i^t\|^2 + 2\|\nabla_{u_i}F_i(u_i^t, v_i^{t+1}, u_0^t)\|^2 \tag{20}$$

Plug (19) and (20) back to (18), we can obatin

$$\mathbb{E}\|u_i^{t+1} - u_i^t\|^2$$

$$\leq \eta_u^2 K_u^2\left(2\sigma_u^2 + \frac{\lambda^2}{2}\sigma_H^2\right) + 2\eta_u^2 K_u^2\|\nabla F(u_i^t, v_i^{t+1}, u_0^t)\|^2 + 2\eta_u^2 K_u\left(2L_{f_u}^2 + \frac{\lambda^2}{2}L_{H_u}^2\right)\sum_{k=0}^{K_u-1}\|u_i^{t,k} - u_i^t\|^2 \tag{21}$$

Then plug (17) and (21) into (15), we can get

$$\mathbb{E}D_2 \leq \frac{1}{M}\sum_{i=1}^{M} -\left(\frac{\eta_u K_u}{2} - \frac{2L_{f_u} + \lambda L_{H_u}}{2}\eta_u^2 K_u^2\right)\|\nabla_{u_i}F(u_i^t, v_i^{t+1}, u_0^t)\|^2$$

$$+ \frac{1}{M}\sum_{i=1}^{M}\left(2L_{f_u}^2 + \frac{\lambda^2}{2}L_{H_u}^2\right)\left(\frac{\eta_u}{2} + \frac{2L_{f_u} + \lambda L_{H_u}}{2}\eta_u^2 K_u\right)\sum_{k=0}^{K_u-1}\|u_i^{t,k} - u_i^t\|^2$$

$$+ \frac{2L_{f_u} + \lambda L_{H_u}}{4}\eta_u^2 K_u^2\left(2\sigma_u^2 + \frac{\lambda^2 \sigma_H^2}{2}\right) \tag{22}$$

The term $\sum_{k=0}^{K_u-1}\|u_i^{t,k}-u_i^t\|^2$ represents the "client drift" in the local SGD steps. We follow Karimireddy et al. (2020) to bound it via Lemma 6. Let $L_1^2 = 2L_{f_u}^2 + \frac{\lambda^2}{2}L_{H_u}^2$, $\sigma_1^2 = 2\sigma_u^2 + \frac{\lambda^2\sigma_H^2}{2}$. When $\eta_u \leq \frac{1}{8L_1K_u}$, by a simple calculation, then we can finally obtain

$$\mathbb{E}D_2 \leq \frac{1}{M}\sum_{i=1}^{M} -\frac{\eta_u K_u}{4}\|\nabla_{u_i}F(u_i^t, v_i^{t+1}, u_0^t)\|^2 + \frac{3}{2}\eta_u^2 K_u^2 L_1 \sigma_1^2$$

$\square$

### B.4 PROOF OF LEMMA 3

The proof of Lemma 3 is very similar to Lemma 2, thus we omit some details here. Through a similar procedure to get (22), we can get

$$\mathbb{E}D_3 \leq \frac{1}{M}\sum_{i=1}^{M} -\left(\frac{\eta_v K_v}{2} - L_{f_v}\eta_v^2 K_v^2\right)\|\nabla_{v_i}f_i(u_i^t, v_i^t)\|^2$$
$$+ \frac{1}{M}\sum_{i=1}^{M}\left(\frac{\eta_v L_{f_v}^2}{2} + L_{f_v}^3\eta_v^2 K_v\right)\sum_{k=0}^{K_v-1}\|v_i^{t,k}-v_i^t\|^2 + \frac{L_{f_v}\eta_v^2 K_v^2\sigma_v^2}{2} \tag{23}$$

Note that the update of $v_i$ is only related to the function $f_i(u_i, v_i)$, not about $H_i(u_i, u_0)$. Thus the formula here is more clean compared to (22). The term $\sum_{k=0}^{K_v-1}\|v_i^{t,k}-v_i^t\|^2$ is also the "client drift" in the local updates of $v_i$. We use Lemma 7 to bound it. When $\eta_v \leq \frac{1}{8L_{f_v}K_v}$, by a simple calculation, we can obtain

$$\mathbb{E}D_3 \leq \frac{1}{M}\sum_{i=1}^{M} -\frac{\eta_v K_v}{4}\|\nabla_{v_i}f_i(u_i^t, v_i^t)\|^2 + \frac{3}{2}\eta_v^2 K_v^2 L_{f_v}\sigma_v^2$$

### B.5 LEMMAS FOR CLIENT DRIFT

The following two lemmas bound the client drift in $u_i$ and $v_i$, respectively. The proofs of two lemmas directly use the Lemma 22 in Pillutla et al. (2022).

**Lemma 6.** *Let $\sigma_1^2 = 2\sigma_u^2 + \frac{\lambda^2\sigma_H^2}{2}$. When $\eta_u \leq \frac{1}{\sqrt{2}K_u L_1}$, the client drift of $u_i$ is bounded by*

$$\mathbb{E}\sum_{k=0}^{K_u-1}\|u_i^{t,k}-u_i^t\|^2 \leq 8K_u^2(K_u-1)\eta_u^2\|\nabla_{u_i}F(u_i^t, v_i^{t+1}, u_0^t)\|^2 + 4(K_u-1)K_u^2\eta_u^2\sigma_1^2 \tag{24}$$

**Lemma 7.** *When $\eta_v \leq \frac{1}{\sqrt{2}K_v L_{f_v}}$, the client drift of $v_i$ is bounded by*

$$\mathbb{E}\sum_{k=0}^{K_v-1}\|v_i^{t,k}-v_i^t\|^2 \leq 8K_v^2(K_v-1)\eta_v^2\|\nabla_{v_i}f_i(u_i^t, v_i^t)\|^2 + 4(K_v-1)K_v^2\eta_v^2\sigma_v^2 \tag{25}$$

### B.6 PROOF OF LEMMA 4

*Proof.*

$$\mathbb{E}\|\nabla_{u_0}F(U^t, u_0^t)\|^2$$
$$\leq 2\mathbb{E}\|\nabla_{u_0}F(U^t, u_0^t) - \nabla_{u_0}F(U^{t+1}, u_0^t)\|^2 + 2\mathbb{E}\|\nabla_{u_0}F(U^{t+1}, u_0^t)\|^2$$
$$= 2\mathbb{E}\|\frac{\lambda}{2M}\sum_{i=1}^{M}\left(\nabla_{u_0}H_i(u_i^t, u_0^t) - \nabla_{u_0}H_i(u_i^{t+1}, u_0^t)\right)\|^2 + 2\mathbb{E}\|\nabla_{u_0}F(U^{t+1}, u_0^t)\|^2$$
$$\leq \frac{\lambda^2}{2M}\sum_{i=1}^{M}\mathbb{E}\|\nabla_{u_0}H_i(u_i^t, u_0^t) - \nabla_{u_0}H_i(u_i^{t+1}, u_0^t)\|^2 + 2\mathbb{E}\|\nabla_{u_0}F(U^{t+1}, u_0^t)\|^2$$
$$\leq \frac{\lambda^2 L_{H_{uu}}^2}{2M}\sum_{i=1}^{M}\mathbb{E}\|u_i^{t+1}-u_i^t\|^2 + 2\mathbb{E}\|\nabla_{u_0}F(U^{t+1}, u_0^t)\|^2 \tag{26}$$

The first term in RHS of above inequality is exactly (21) in the above proof. Plug (21) into (26), we can get

$$\mathbb{E}\|\nabla_{u_0}F(U^t,u_0^t)\|^2$$

$$\leq \frac{\lambda^2 L_{H_{uu}}^2}{2M}\sum_{i=1}^{M}\left(\eta_u^2 K_u^2 \sigma_1^2 + 2\eta_u^2 K_u^2\|\nabla F(u_i^t,v_i^{t+1},u_0^t)\|^2 + 2\eta_u^2 K_u L_1^2\sum_{k=0}^{K_u-1}\|u_i^{t,k}-u_i^t\|^2\right)$$

$$+2\mathbb{E}\|\nabla_{u_0}F(U^{t+1},u_0^t)\|^2$$

Again we use the Lemma 6 to bound the client drift. When $\eta_u \leq \frac{1}{8L_1 K_u}$, plugging the Lemma 6 into the above inequality, and by a calculation to simplify the coefficient, we can finally get

$$\mathbb{E}\|\nabla_{u_0}F(U^t,u_0^t)\|^2$$

$$\leq \frac{6}{5}\lambda^2 L_{H_{uu}}^2 \eta_u^2 K_u^2 \frac{1}{M}\sum_{i=1}^{M}\|\nabla_{u_i}F_i(u_i^t,v_i^{t+1},u_0^t)\|^2 + \frac{3}{5}\lambda^2 L_{H_{uu}}^2 \eta_u^2 K_u^2 \sigma_1^2 + 2\|\nabla_{u_0}F(U^{t+1},u_0^t)\|^2$$

$\square$

The proof of Lemma 7 follows the same procedure of the proof of Lemma 6, thus omitted here.

## C    FedReCo with Partial Variance Reduction

### C.1    Partial Variance Reduction

At each round, we calculate the stochastic gradients for regularization function $H_i(u_i,u_0)$. To reduce the effect of variance of the stochastic gradients, we can use variance reduction techniques in the process of training. However, due to the ineffectiveness of variance reduction in the complicated non-convex functions, especially in neural network training, we wish to remain the randomness brought by the stochastic gradients of function $f_i(u_i,v_i)$. Thus we propose a partial variance reduction method for the stochastic regularization term.

Specially, at the end of $t$-th round, the client $i$ calculates a full gradient of $H_i(u_i^t,u_0^{t-1})$ with respect to $u_i$:

$$g_i^t = \nabla_{u_i}H_i(u_i^t,u_0^{t-1})$$

and a full gradient with respect to $u_0$: $\nabla_{u_0}H_i(u_i^t,u_0^{t-1})$. Then the client remains $g_i^t$ and sends the full gradient $\nabla_{u_0}H_i(u_i^t,u_0^{t-1})$ to the server. The server aggregates the full gradient from all the clients and updates the $u_0$. Thus $u_0$ is updated via full gradient descent, not SGD. At the next round, the server broadcast $u_0$ to all the clients. The $v_i$ is updated as the same local SGD manner. But the $u_i$ is updated as

$$u_i^{t,k+1} = u_i^{t,k} - \eta_u\left[\tilde{\nabla}_{u_i}f_i(v_i^{t+1},u_i^{t,k}) + \frac{\lambda}{2}\left(\tilde{\nabla}_{u_i}H_i(u_i^{t,k},u_0^t) - \tilde{\nabla}_{u_i}H_i(u_i^t,u_0^{t-1}) + g_i^t\right)\right]$$

where $G_i^{t,k} = \tilde{\nabla}_{u_i}H_i(u_i^{t,k},u_0^t) - \tilde{\nabla}_{u_i}H_i(u_i^t,u_0^{t-1}) + g_i^t$ is an approximated full gradient of $\nabla_{u_i}H_i(u_i^{t,k},u_0^t)$. The detailed algorithm is shown in Algorithm 2.

Without partial variance reduction, the variance of stochastic gradient $\tilde{\nabla}_{u_i}H_i(u_i^{t,k},u_0^t)$ is

$$\|\tilde{\nabla}_{u_i}H_i(u_i^{t,k},u_0^t) - \nabla H_i(u_i^{t,k},u_0^t)\|^2 \leq \sigma_H^2$$

When we use the partial variance reduction as above, the variance is

$$\|\tilde{\nabla}_{u_i}H_i(u_i^{t,k},u_0^t) - \tilde{\nabla}_{u_i}H_i(u_i^t,u_0^{t-1}) + g_i^t - \nabla H_i(u_i^{t,k},u_0^t)\|^2$$

$$=\|\left(\tilde{\nabla}_{u_i}H_i(u_i^{t,k},u_0^t) - \tilde{\nabla}_{u_i}H_i(u_i^t,u_0^{t-1})\right) - \left(\nabla H_i(u_i^{t,k},u_0^t) - g_i^t\right)\|^2$$

$$\leq\|\tilde{\nabla}_{u_i}H_i(u_i^{t,k},u_0^t) - \tilde{\nabla}_{u_i}H_i(u_i^t,u_0^{t-1})\|^2$$

$$=\|\tilde{\nabla}_{u_i}H_i(u_i^{t,k},u_0^t) - \tilde{\nabla}_{u_i}H_i(u_i^{t,k},u_0^t) + \tilde{\nabla}_{u_i}H_i(u_i^{t,k},u_0^t) - \tilde{\nabla}_{u_i}H_i(u_i^t,u_0^{t-1})\|^2$$

$$\leq 2L_{H_u}^2\|u_i^{t,k}-u_i^t\|^2 + 2L_{H_{uu}}^2\|u_0^t-u_0^{t-1}\|^2$$

where the first inequality is from the fact $\mathbb{E}\|x - \mathbb{E}x\|^2 \leq E\|x\|^2$, and the last inequality if from the smoothness properties. The variance is bounded by the difference of $u_i$ and $u_0$ between two iterations. Note that $\sum_{k=0}^{K_u-1}u_i^{t,k}-u_i^t$ is the client drift in each client due to data heterogeneity. We can incorporate it in the client drift term. Since we update the variables alternatively, the variance is also bounded by $\|u_0^t-u_0^{t-1}\|^2$. We can use telescopic cancellation to handle it.

---
**Algorithm 2** FedReCo with partial variance reduction

---
**Input**: Step size $\eta_u, \eta_v, \eta_0$, penalty parameter $\lambda$
**Initialize**: Initialize $u_0^0$ for server, initialize $u_i$ and $v_i$ for $i$-th client

 1: **for** $t = 0, 1, ..., T-1$ **do**
 2:     **Server**:
 3:     Broadcast $u_0^t$ to all clients
 4:     Receive full gradient $\nabla_{u_0} H_i(u_i^{t+1}, u_0^t)$ from all the clients
 5:     Update $u_0^{t+1}$: $u_0^{t+1} = u_0^t - \eta_0 \frac{1}{M} \sum_{i=1}^{M} \nabla_{u_0} H_i(u_i^{t+1}, u_0^t)$
 6:     **client** $i$:
 7:     Receive $u_0^t$ from master node, let $u_i^{t,0} = u_i^t$
 8:     **for** $k = 0, 1, ..., K_v - 1$ **do**
 9:       Randomly select one (batch of) sample, calculate stochastic gradients $\tilde{\nabla}_{v_i} f_i(v_i^{t,k}, u_i^t)$
10:       Update $v_i^{t,k+1} = v_i^{t,k} - \eta_v \tilde{\nabla}_{v_i} f_i(v_i^{t,k}, u_i^t)$
11:     **end for**
12:     Let $v_i^{t+1} = v_i^{t,K_v}$
13:     **for** $k = 0, 1, ..., K_u - 1$ **do**
14:       Randomly select one (batch of) sample, calculate stochastic gradients $\tilde{\nabla}_{u_i} f_i(v_i^{t+1}, u_i^{t,k})$ and
        $\tilde{\nabla}_{u_i} H_i(u_i^{t,k}, u_0^t)$
15:       Update $u_i^{t,k+1} = u_i^{t,k} - \eta_u \left[ \tilde{\nabla}_{u_i} f_i(v_i^{t+1}, u_i^{t,k}) + \frac{\lambda}{2} \left( \tilde{\nabla}_{u_i} H_i(u_i^{t,k}, u_0^t) - \tilde{\nabla}_{u_i} H_i(u_i^t, u_0^{t-1}) + g_i^t \right) \right]$
16:     **end for**
17:     Let $u_i^{t+1} = u_i^{t,K_u}$
18:     Calculate full gradient $g_i^{t+1} = \nabla_{u_i} H_i(u_i^{t+1}, u_0^t)$ and $\nabla_{u_0} H_i(u_i^{t+1}, u_0^t)$
19:     Send full gradient $\nabla_{u_0} H_i(u_i^{t+1}, u_0^t)$ to the server
20: **end for**

---

## C.2    Convergence Analysis

**Theorem 2** (Convergence of FedReCo-PVR). *Suppose that Assumptions 1 and 2 hold. Let* $L_1^{'2} = 4L_{f_u}^2 + 2\lambda^2 L_{H_u}^2$. *When learning rates satisfy* $\eta_0 = \frac{\eta}{L_{H_u}}$, $\eta_u = \frac{\eta}{L_1' K_u}$, $\eta_v = \frac{\eta}{L_{f_v} K_v}$, *and $\eta$ is chosen on the parameters* $\lambda, L_{H_u}, L_1, L_{f_v}, L_{H_{uu}}, L_{f_{uv}}, \sigma_H, \sigma_u, \sigma_v$, *then ignoring absolute constants, we have:*

$$\frac{1}{T} \sum_{t=0}^{T-1} \left( \frac{1}{8L_{H_u}} \Gamma_1^t + \frac{1}{16L_1'} \Gamma_2^t + \frac{1}{8L_{f_v}} \Gamma_3^t \right) \lesssim \frac{\Sigma_1^{\frac{1}{2}}}{\sqrt{T}} + \frac{\Sigma_2^{\frac{1}{3}}}{T^{\frac{2}{3}}} + O\left( \frac{1}{T} \right) \tag{27}$$

*where*

$$\Sigma_1 = \frac{5}{2} \frac{\sigma_u^2}{L_1'} + \frac{3}{2} \frac{\sigma_v^2}{L_{f_v}}, \quad \Sigma_2 = \left( \frac{\lambda^2}{64} \frac{L_{H_{uu}}^2}{L_{H_u} L_1'^2} \sigma_u^2 + \frac{3}{20} \frac{L_{f_{uv}}^2}{L_1' L_{f_v}^2} \sigma_v^2 \right)$$

*are positive constants depending on Lipschitz constants and stochastic variance.*

We can see in the $\Sigma_1$ and $\Sigma_2$, there is no $\sigma_H^2$ compared to the FedReCo algorithm without variance reduction. We remove the impact of additional noise brought by regularization term theoretically. However, we have observed that PVR almost brings no improvement in the practical training of neural network models. And FedReCo algorithm is robust enough to the additional noise.

## D    Proof of FedReCo-PVR

### D.1    Proof Outline of Theorem 2

We provide a proof outline of Theorem 2 here and omit some proofs of technical lemmas.

Similarly, we can break the difference of cost function $F$ as (10). Then we can obtain the lemmas to bound $D_1, D_2$. Note that $D_3$ is the same as FedReCo without partial variance reduction.

The $u_0$ is now update by full gradient descent, not SGD. Thus we have the following Lemma for $D_1$.

**Lemma 8.** *The expectation of $D_1$ satisfies*

$$\mathbb{E}D_1 \leq -\eta_0\left(1-\frac{\lambda L_{H_u}\eta_0}{4}\right)\left\|\frac{1}{M}\frac{\lambda}{2}\sum_{i=1}^M \nabla_u H_i(u_i^{t+1},u_0^t)\right\|^2 \tag{28}$$

*When $\eta_0 \leq \frac{2}{\lambda L_{H_u}}$, the expectation is*

$$\mathbb{E}D_1 \leq -\frac{\eta_0}{2}\|\nabla_{u_0}F(U^{t+1},u_0^t)\|^2 \tag{29}$$

**Lemma 9.** *Let $L_1'^2 = 4L_{f_u}^2 + 2\lambda^2 L_{H_u}^2$. When $\eta_u \leq \frac{1}{16L_1'K_u}$, the expectation of $D_2$ is*

$$\mathbb{E}D_2 \leq \frac{1}{M}\sum_{i=1}^M -\frac{\eta_u K_u}{4}\|\nabla_{u_i}F(u_i^t,v_i^{t+1},u_0^t)\|^2 + \frac{5}{2}\eta_u^2 K_u^2 L_1'\sigma_u^2 + \frac{5}{4}\lambda^2\eta_u^2 K_u^2 L_1'L_{H_{uu}}^2\|u_0^t-u_0^{t-1}\|^2 \tag{30}$$

With the two lemmas and Lemma 3 in previous sections, we can write

$$\frac{\eta_0}{2}\mathbb{E}\|\nabla_{u_0}F(U^{t+1},u_0^t)\|^2 + \frac{1}{M}\sum_{i=1}^M\frac{\eta_u K_u}{4}\mathbb{E}\|\nabla_{u_i}F_i(u_i^t,v_i^{t+1},u_0^t)\|^2$$

$$+\frac{1}{M}\sum_{i=1}^M\frac{\eta_v K_v}{4}\mathbb{E}\|\nabla_{v_i}f_i(u_i^t,v_i^t)\|^2$$

$$\leq\mathbb{E}\left[F(u_i^t,v_i^t,u_0^t)-F(u_i^{t+1},v_i^{t+1},u_0^{t+1})\right]+\frac{5}{2}\eta_u^2 K_u^2 L_1'\sigma_u^2+\frac{3}{2}\eta_v^2 K_v^2 L_{f_v}\sigma_v^2+\frac{5}{4}\lambda^2\eta_u^2 K_u^2 L_1'^2 L_{H_{uu}}^2\|u_0^t-u_0^{t-1}\|^2 \tag{31}$$

Note that the metric we use to measure the convergence is $\|\nabla_{u_0}F(U^t,u_0^t)\|^2$ and $\frac{1}{M}\|\nabla_{u_i}F_i(u_i^t,v_i^t,u_0^t)\|^2$, thus we have the following lemma to measure the difference.

**Lemma 10.** *The expectation of $\|\nabla_{u_0}F(U^t,u_0^t)\|^2$ is bounded by*

$$\mathbb{E}\|\nabla_{u_0}F(U^t,u_0^t)\|^2$$

$$\leq\frac{9}{8}\lambda^2 L_{H_{uu}}^2\eta_u^2 K_u^2\frac{1}{M}\sum_{i=1}^M\|\nabla_{u_i}F_i(u_i^t,v_i^{t+1},u_0^t)\|^2+\frac{1}{64}\lambda^2 L_{H_{uu}}^2\eta_u^2 K_u^2\sigma_u^2$$

$$+\frac{5}{8}\lambda^4\eta_u^2 K_u^2 L_{H_{uu}}^4\|u_0^t-u_0^{t-1}\|^2+2\|\nabla_{u_0}F(U^{t+1},u_0^t)\|^2 \tag{32}$$

Using the Lemma 10 and Lemma 5 in previous sections, we can get

$$\frac{\eta_0}{8}\|\nabla_{u_0}F(U^t,u_0^t)\|^2+\frac{\eta_u K_u}{16}\frac{1}{M}\sum_{i=1}^M\|F_i(u_i^t,v_i^t,u_0^t)\|^2+\frac{\eta_v K_v}{8}\frac{1}{M}\sum_{i=1}^M\|\nabla_{v_i}f_i(u_i^t,v_i^t)\|^2$$

$$\leq\frac{\eta_0}{4}\|\nabla_{u_0}F(U^{t+1},u_0^t)\|^2+\left(\frac{9}{64}\lambda^2\eta_0\eta_u^2 L_{H_{uu}}^2 K_u^2+\frac{1}{8}\eta_u K_u\right)\frac{1}{M}\sum_{i=1}^M\|\nabla_{u_i}F_i(u_i^t,v_i^{t+1},u_0^t)\|^2$$

$$+\left(\frac{3}{10}L_{f_{uv}}^2\eta_u K_u\eta_v^2 K_v^2+\frac{1}{8}\eta_v K_v\right)\frac{1}{M}\sum_{i=1}^M\|\nabla_{v_i}f_i(u_i^t,v_i^t)\|^2$$

$$+\frac{1}{64}\lambda^2 L_{H_{uu}}^2\eta_0\eta_u^2 K_u^2\sigma_u^2+\frac{3}{20}L_{f_{uv}}^2\eta_u K_u\eta_v^2 K_v^2\sigma_v^2+\frac{5}{64}\lambda^4\eta_u^2\eta_0 K_u^2 L_{H_{uu}}^4\|u_0-u_0^{t-1}\|^2 \tag{33}$$

When $\lambda^2 \eta_0 \eta_u K_u L^2_{H_{uu}} \leq \frac{8}{9}$ and $\eta_u K_u \eta_v K_v L^2_{f_{uv}} \leq \frac{5}{12}$, we have

$$
\frac{\eta_0}{8} \|\nabla_{u_0} F(U^t, u_0^t)\|^2 + \frac{\eta_u K_u}{16} \frac{1}{M} \sum_{i=1}^M \|F_i(u_i^t, v_i^t, u_0^t)\|^2 + \frac{\eta_v K_v}{8} \frac{1}{M} \sum_{i=1}^M \|\nabla_{v_i} f_i(u_i^t, v_i^t)\|^2
$$

$$
\leq \frac{\eta_0}{2} \mathbb{E} \|\nabla_{u_0} F(U^{t+1}, u_0^t)\|^2 + \frac{1}{M} \sum_{i=1}^M \frac{\eta_u K_u}{4} \mathbb{E} \|\nabla_{u_i} F_i(u_i^t, v_i^{t+1}, u_0^t)\|^2
$$

$$
+ \frac{1}{M} \sum_{i=1}^M \frac{\eta_v K_v}{4} \mathbb{E} \|\nabla_{v_i} f_i(u_i^t, v_i^t)\|^2 - \frac{\eta_0}{4} \|\nabla_{u_0} F(U^{t+1}, u_0^t)\|^2
$$

$$
+ \frac{1}{64} \lambda^2 L^2_{H_{uu}} \eta_0 \eta_u^2 K_u^2 \sigma_u^2 + \frac{3}{20} L^2_{f_{uv}} \eta_u K_u \eta_v^2 K_v^2 \sigma_v^2 + \frac{5}{64} \lambda^4 \eta_u^2 \eta_0 K_u^2 L^4_{H_{uu}} \|u_0 - u_0^{t-1}\|^2
$$

$$
\leq \mathbb{E} \big[ F(U^t, V^t, u_0^t) - F(U^{t+1}, V^{t+1}, u_0^{t+1}) \big] - \frac{\eta_0}{4} \|\nabla_{u_0} F(U^{t+1}, u_0^t)\|^2
$$

$$
+ \left( \frac{5}{4} \lambda^2 \eta_u^2 K_u^2 L_1' L^2_{H_{uu}} + \frac{5}{64} \lambda^4 \eta_u^2 \eta_0 K_u^2 L^4_{H_{uu}} \right) \|u_0^t - u_0^{t-1}\|^2
$$

$$
+ \left( \frac{5}{2} \eta_u^2 K_u^2 L_1' + \frac{1}{64} \lambda^2 L^2_{H_{uu}} \eta_0 \eta_u^2 K_u^2 \right) \sigma_u^2 + \left( \frac{3}{2} \eta_v^2 K_v^2 L_{f_v} + \frac{3}{20} L^2_{f_{uv}} \eta_u K_u \eta_v^2 K_v^2 \right) \sigma_v^2 \tag{34}
$$

Let $\eta_0 = \frac{\eta}{L_{H_u}}$, $\eta_u = \frac{\eta}{L_1' K_u}$, $\eta_v = \frac{\eta}{L_{f_v} K_v}$. We can obtain

$$
\frac{1}{8 L_{H_u}} \mathbb{E} \|\nabla_{u_0} F(U^t, u_0^t)\|^2 + \frac{1}{16 L_1'} \frac{1}{M} \sum_{i=1}^M \|\nabla_{u_i} F_i(u_i^t, v_i^t, u_0^t)\|^2 + \frac{1}{8 L_{f_v}} \frac{1}{M} \sum_{i=1}^M \|\nabla_{v_i} f_i(u_i^t, u_i^t)\|^2
$$

$$
\leq \frac{\mathbb{E} \big[ F(u_i^t, v_i^t, u_0^t) - F(u_i^{t+1}, v_i^{t+1}, u_0^{t+1}) \big]}{\eta} - \frac{\eta_0}{4\eta} \|\nabla_{u_0} F(U^{t+1}, u_0^t)\|^2 + \frac{\eta_0}{4\eta} \|\nabla_{u_0} F(U^t, u_0^{t-1})\|^2
$$

$$
+ \eta \left( \frac{5}{2} \frac{\sigma_u^2}{L_1'} + \frac{3}{2} \frac{\sigma_v^2}{L_{f_v}} \right) + \eta^2 \left( \frac{\lambda^2}{64} \frac{L^2_{H_{uu}}}{L_{H_u} L_1'^2} \sigma_u^2 + \frac{3}{20} \frac{L^2_{f_{uv}}}{L_1' L^2_{f_v}} \sigma_v^2 \right)
$$

$$
= \frac{\mathbb{E} \big[ F(u_i^t, v_i^t, u_0^t) - F(u_i^{t+1}, v_i^{t+1}, u_0^{t+1}) \big]}{\eta} + \frac{1}{4 L_{H_u}} \big( \|\nabla_{u_0} F(U^t, u_0^{t-1})\|^2 - \|\nabla_{u_0} F(U^{t+1}, u_0^t)\|^2 \big)
$$

$$
+ \eta \left( \frac{5}{2} \frac{\sigma_u^2}{L_1'} + \frac{3}{2} \frac{\sigma_v^2}{L_{f_v}} \right) + \eta^2 \left( \frac{\lambda^2}{64} \frac{L^2_{H_{uu}}}{L_{H_u} L_1'^2} \sigma_u^2 + \frac{3}{20} \frac{L^2_{f_{uv}}}{L_1' L^2_{f_v}} \sigma_v^2 \right) \tag{35}
$$

Define $\Sigma_1 = \frac{5}{2} \frac{\sigma_u^2}{L_1'} + \frac{3}{2} \frac{\sigma_v^2}{L_{f_v}}$ and $\Sigma_2 = \left( \frac{\lambda^2}{64} \frac{L^2_{H_{uu}}}{L_{H_u} L_1'^2} \sigma_u^2 + \frac{3}{20} \frac{L^2_{f_{uv}}}{L_1' L^2_{f_v}} \sigma_v^2 \right)$. Applying telescopic cancellation through $t=0$ to $t=T-1$, we have

$$
\frac{1}{T} \sum_{t=0}^{T-1} \left( \frac{1}{8 L_{H_u}} \Gamma_1^t + \frac{1}{16 L_1'} \Gamma_2^t + \frac{1}{8 L_{f_v}} \Gamma_3^t \right)
$$

$$
\leq \frac{\big[ F(U^0, V^0, u_0^0) - F(U^T, V^T, u_0^T) \big]}{\eta T} + \frac{1}{4 L_{H_u} T} \big( \|\nabla_{u_0} F(U^1, u_0^0)\|^2 - \|\nabla_{u_0} F(U^T, u_0^{T-1})\|^2 \big)
$$

$$
+ \eta \Sigma_1 + \eta^2 \Sigma_2
$$

$$
\leq \frac{F(U^0, V^0, u_0^0) - F_{\min}}{\eta T} + \frac{1}{4 L_{H_u} T} \|\nabla_{u_0} F(U^1, u_0^0)\|^2 + \eta \Sigma_1 + \eta^2 \Sigma_2 \tag{36}
$$

We need to make $\eta$ satisfy the conditions in all the above proofs, thus

$$
\eta \leq \min \left\{ \frac{2}{\lambda}, \quad \frac{1}{16}, \quad \sqrt{\frac{8 L_{H_u} L_1'}{9 \lambda^2 L^2_{H_{uu}}}}, \quad \sqrt{\frac{5 L_{f_v} L_1'}{12 L^2_{f_{uv}}}} \right\}
$$

Define $\Delta F = F(u_i^0, v_i^0, u_0^0) - F_{\min}$. Let

$$\eta = \frac{1}{\frac{\lambda}{2} + 16 + \sqrt{\frac{9\lambda^2 L_{H_{uu}}^2}{8 L_{H_u} L_1'}} + \sqrt{\frac{12 L_{f_{uv}}^2}{5 L_{f_v} L_1'}} + \sqrt{\Sigma_1 T} + \Sigma_2^{\frac{1}{3}} T^{\frac{1}{3}}}$$

Then we can get

$$\frac{1}{T}\sum_{t=0}^{T-1}\left(\frac{1}{8 L_{H_u}}\Gamma_1^t + \frac{1}{16 L_1'}\Gamma_2^t + \frac{1}{8 L_{f_v}}\Gamma_3^t\right)$$

$$\leq \frac{(\Delta F + 1)\sqrt{\Sigma_1}}{\sqrt{T}} + \frac{(\Delta F + 1)\Sigma_2^{\frac{1}{3}}}{T^{\frac{2}{3}}} + \frac{1}{T}\left[\Delta F\left(\frac{\lambda}{2} + 8 + \sqrt{\frac{12\lambda^2 L_{H_{uu}}^2}{5 L_{H_u} L_1}} + \sqrt{\frac{12 L_{f_{uv}}^2}{5 L_{f_v} L_1}}\right) + \frac{\|\nabla_{u_0} F(U^1, u_0^0)\|^2}{4 L_{H_u}}\right]$$

$$(37)$$

Ignoring absolute constants, we have

$$\frac{1}{T}\sum_{t=0}^{T-1}\left(\frac{1}{8 L_{H_u}}\Gamma_1^t + \frac{1}{16 L_1'}\Gamma_2^t + \frac{1}{8 L_{f_v}}\Gamma_3^t\right) \lesssim \frac{\Sigma_1^{\frac{1}{2}}}{\sqrt{T}} + \frac{\Sigma_2^{\frac{1}{3}}}{T^{\frac{2}{3}}} + O\left(\frac{1}{T}\right) \tag{38}$$