# OpenReview forum: "Consensus Optimization at Representation: Improving Personalized Federated Learning via Data-Centric Regularization"
_ICLR.cc/2024/Conference — ICLR 2024 Conference Withdrawn Submission_

### Official Review · Reviewer_QkUY · 2023-10-14

**Soundness:** 2 fair
**Presentation:** 2 fair
**Contribution:** 2 fair
**Rating:** 3
**Confidence:** 4

**Summary:**

This paper proposes a pFL method FedReCo which adopts the representation consensus to optimize the personalized models. It also provides a theoretical analysis of the convergence. Experiments on two datasets with a single setup respectively are conducted to validate the performance of the proposed FedReCo.

**Strengths:**

1. The writing of this paper is easy to understand.
2. The idea of adopting the representation consensus in pFL is interesting.

**Weaknesses:**

1. This work lacks novelty and does not cite a similar work [R1]. The author claims the cited [R2] is the most related work to this paper. However, [R2] solves the issue through a data-driven aggregation method. There is no similar idea in the proposed FedReCo to adopt the data properties. In [R1] the method for representation consensus in FL has been studied and this paper does not provide a systematic comparison to clarify improvement points and significant contributions between the proposed FedReCo and the [R1] work. Except for the variant of variance reduction adopted in the classifier, the current version looks like the global contrastive training loss in [R1].

   **[R1]**: FedBR: Improving Federated Learning on Heterogeneous Data via Local Learning Bias Reduction **(ICML 2023)**

   **[R2]**: FedCR: Personalized Federated Learning Based on Across-Client Common Representation with Conditional Mutual Information Regularization **(ICML 2023)**

2. Proof is incremental work without the analysis of $M$ and local epochs. The current analysis does not indicate the optimal convergence rates. General FedAvg has shown the convergence achieves $\mathcal{O}(\frac{1}{\sqrt{MKT}})$ where $K$ is the local interval [R3]. However, the analysis for FedReCo only indicates a $\mathcal{O}(\frac{1}{\sqrt{T}})$ rate.

    **[R3]**: Achieving Linear Speedup with Partial Worker Participation in non-iid Federated Learning (ICLR2021)

3. The proposed FedReCo seems to not support partial participation. It may be a large limitation on the practical applications with large-scale edge devices.
4. Experimental setups are limited. Section 6.1 only shows that the experimental setups are a total of 50 clients with the pathological dataset split of 2 classes per client on CIFAR and 4 classes per client on FMNIST. This makes the experimental data reported in this paper unreliable enough. The general setups include changes in the total number of clients $M$, changes in the degree of heterogeneity, and changes in local training epochs. Through extensive comparisons, we can validate the true effectiveness of a proposed algorithm. The experimental settings of this paper are too monotonous, making the experimental results only seem to reflect certain advantages. Whether the algorithm has an overall lead is unknown.

My questions are stated in the following.

**Questions:**

1. In the Introduction section, the sentence “At each iteration, a local (stochastic) gradient … … to the server for a global update.” (Below Eq.(1)) is required to be corrected. FL usually does not set the local interval as 1 like the general distributed training and it does not transfer the local information so frequently since the limited communication bottleneck.
2. Figure 2 does not seem to fully reflect the issue of similarity in the entire training. I only can see, at the beginning of the training, i.e. t=1 and 10, that there is a dissimilarity issue between local and global parameters. However, this difference is reasonable at the beginning of training because all local clients yet do not search for a valid local minimum. In other words, everyone will not reach a high consensus solution at $t=10$. It has been theoretically proven that the consensus in FL will increase as the communication $t$ increases. Furthermore, Figure.2 clearly shows that the similarity significantly increases as $t$ increases, which is consistent with the theory. So I would like to know whether this inconsistency phenomenon still exists in the later stages of training and after converging, and how it will damage the final performance in pFL.
3. In section 3.2, $h_{ij}(u)$ is the mapping function that maps the input $x$ to a intermediate representation. So what is the calculation of $h(x|u)$? Is it the feature extractor itself? I did not find a specific definition of this mapping calculation in the context. As I know, some previous studies use linear projection and some use a complicated network to align the feature information. How is the mapping function in this work calculated?
4. (1) On Lines 4 and 5 in Algorithm 1, the index of $\nabla H_i(u_i^{t+1},u_0^t)$ seems to should be corrected as $\nabla H_i(u_i^{t},u_0^{t-1})$ since its calculation is stated in Line 18. All the indices are required to be checked again. (2) On Line 14, "pass the same batch of t samples", it looks like a typo or what is the meaning of t?
5. An interesting update rule is that in Line 5 in Algorithm 1, the global feature extractor $u_0^t$ is updated by the local stochastic gradient $\nabla H_i(u_i^{t+1},u_0^t)$. From the perspective of optimization, the most efficient update here seems to be directly minimizing the regularization term, i.e.,  $\min_{u_0} H$. Then the global $u_0$ could be solved by the average of the local $u_i$. However, FedReCo adopts the one-step stochastic gradient descent to update global $u$. This confuses me a bit. What is the motivation behind this update?
6. The proposed FedReCo is a little similar to the [R1]. Both of them adopt the consensus representation between the local and global parameters. If the additional fake data generation in [R1] is removed, FedReCo is like its simplified version. What are the improvements and novelty of this work compared with [R1]?

    **[R1]**: FedBR: Improving Federated Learning on Heterogeneous Data via Local Learning Bias Reduction (ICML 2023)

7. Assumption 2 seems to be missing an explanation of the unbiasedness of stochastic gradients.
8. The author should at least try to increase the total number of clients $M$ and change the local heterogeneity levels to comprehensively compare the performance among benchmarks and the proposed method.

Thanks for the authors and I will re-score according to the response.

---

### Official Review · Reviewer_qMXj · 2023-10-29

**Soundness:** 4 excellent
**Presentation:** 3 good
**Contribution:** 2 fair
**Rating:** 5
**Confidence:** 3

**Summary:**

This paper introduces a new federated learning method, called FedReCo, that aims at handling heterogeneity by enforcing consensus but at representation level.
The authors propose a to achieve this by formulating the model as an encoder and a decoder and by solving a regularized problem that enforce the output of the encoders to be closed to each other.
The paper propose a non-convex convergence result based on standard Lipschitz-gradient and bounded noise assumptions.
Then experiments are conducted on FMNIST and CIFAR10 to compare the performance of FedReCo versus other standard and personalized FL methods.

**Strengths:**

- Good introduction and explanation of the motivation: decoupling local models into similar (via regularization), but not exactly the same, encoders but fully personalized predictors
- Strong theoretical result supporting the proposed method
- Experimental results showing the efficiency of the method and the application of differential privacy to it

**Weaknesses:**

- FedReCo algorithm presented in Algorithm 1 requires more than 2 times more gradient computations than standard (iterations in v, in u and to compute the gradient of the regularizer)
- the formulation proposed in the paper does not capture the sequential structure of deep neural networks
- Experiments are given in time but not in epochs which might be a weakness of the algorithm compare to methods requiring less local computations. I disagree with the statement in page 4, section 4.2 : "The local training burden does increase too much compared to FedAvg". A fair experiment should show the opposite.
- The methods chosen for comparison in section 6.1 are not compared to FedReCo
- Partial variance reduction for $H_i (u_i, u_0)$is introduced, justified even though VR is not know to improve in deep learning applications, then appears to be inefficient

**Questions:**

**Questions**
1) What are the main differences between enforcing
2) As the focus is to share the same common feature extractor, did you consider the multi-task learning (different structure for the predictors) framework where it makes sense only to share them? Would FedReCo work in this framework?
3) Page 2, "distributed stochastic gradient descent" -> isn't it local-SGD [Stich "Local SGD converges fast and communicates little.", 2018] ?
4) Sec. 3.1, bottom of page 4, "slightly dissimilar" : according to Figure 2, the layers are only dissimilar at the layer 17 (out of 18). The decrease wrt to the depth of the network is not blatant. I think that running the same experiments in Figure 2 for IID setting and deeper networks might be a way to show the influence of client drift on layer (dis)similarity. What do the authors think?
5) Why is there a subscript $j$ in $h_{ij} (x_{ij} | u)$ in sec 3.2 ? How does the mapping from a data sample to the intermediate distribution depends on $i$ and $j$ ? For me it should be removed and maybe one can write: $h_{ij} (u) := h (x_{ij} | u)$
6) Is $L_1$ an upper bound on the Lipschitz constant of $\nabla_{u_i} F (u_i, v_i, u_0)$ ?
7) Where is the code? Did I miss it or its link?
8) Table 3: I have checked the Appendix A3, but could you explain why it is fair to compare DP applied at representation level for FedReCo and at the output of the network for FedAvg ?

**Comments**
1) Citations are sometimes badly integrated into sentences and make reading difficult, eg section 6.1, related works
2) Figure 1: make the fonts and the figure bigger, not readable as it is
3) Cite gradient inversion risk [Zhu, "Deep Leakage from Gradients", 2019] on page 3, sharing gradients is not necessarily more private than models
4) Inverted arguments in eq (5) $ F (u_0, U)$ and in the definition of $\Gamma^t_1$ and $\Gamma^t_2$ on page 7
5) page 8, "In the $\Sigma_1 at ...$ -> to reformulate


**Typo**
1) "differential private variant" -> maybe "differentially private variant" ?
2) Page 3: " ... Sec. 6.2)"
3) Page 7: "... since our method is fully personalized" ?
4) Page 7: in the definition of $\Gamma^t_3$ the second argument should be $v^t_i$

---

### Official Review · Reviewer_EJeM · 2023-11-01

**Soundness:** 2 fair
**Presentation:** 3 good
**Contribution:** 2 fair
**Rating:** 3
**Confidence:** 4

**Summary:**

This work proposes a personalized federated learning procedure where each client learns its personal model but the representations learned at each client are regularized to be close to each other. The paper gives an algorithm to optimize this objective and analyzes its convergence in the non-convex case assuming the participation of all the clients. Finally, the authors also present a variant of their method with local differential privacy, showing that it gives better performance than a naive baseline (FedAvg + fine-tune) in the experiments.

**Strengths:**

The paper is clear and does a good job of conveying the main ideas. The math appears to be right -- I did some spot checks although I did not verify the proofs line-by-line. The proposed approach is more strongly compatible with differential privacy.

**Weaknesses:**

**Originality**: The novelty of the paper is limited as it is a combination of two well-explored ideas in personalized federated learning: (a) consensus formulations (pFedMe, Ditto), and (b) separating the feature learning + prediction head (Arivazaghan et al, Collins et al. Pillutla et al.). The robustness to noise (for differential privacy) is interesting but not well enough explored empirically or in theory (more on this later). The convergence analysis also appears to be a straightforward extension of previous works ([Hanzely et al.](https://arxiv.org/abs/2102.09743), Pillutla et al.).

**Technical Depth**:
- A key technical challenge in prior work on personalized FL has been to address partial client participation. The paper does not tackle this question and assumes full participation of clients for the analysis (and even the experiments, as far as I can tell). This makes the proof techniques more mechanical as they are based on established techniques.
- The smoothness and bounded variance assumptions made on the regularization $H_i$ are not interpretable. Precisely, what properties of the feature mapping $h_{ij}(x)$ do you need so that the resulting regularization term $H_i$ is smooth and has bounded variance?

**Experiments**:
- The experimental evaluations are limited to settings with a small number of clients and a large number of examples per client. It might be interesting to perform an experimental comparison in settings where there is a limited amount of data per client (e.g. around 50 or 100 examples per client).
- Given the above, an important baseline is to train an individual model per client with no communication.
- The experiments must also report the performance of the global feature representation $u_0$ e.g. by linear probing (i.e. just training the classification head).
- At the top of page 9, we have a statement that reads "FedReCo communicates less than Ditto". It would be interesting to normalize these comparisons by the number of bytes transmitted to drive this point home.
- It is hard to gauge the significance of the results without error bars from multiple repetitions. It would be good to have those.


**Results with DP**: The results of FedReCo under extremely small levels of privacy loss appear too good to be true at first glance. Additional context and details are necessary to understand the significance of these results.
- The non-private baseline for FMNIST (93.09%) is worse than the private ones (93.14%). This is quite counterintuitive. At a minimum, error bars are necessary to contextualize the results.
- The reported $\varepsilon$ values are surprisingly small (especially given that this is local DP and not distributed DP, as is common in FL). Is the reported $\varepsilon$ per round or the cumulative $\varepsilon$ at the end of training (i.e. using composition)? Is there any privacy amplification by sampling? Specifying those details in the appendix would be good.
- A clip norm of 10 was used for both FedReCo and FedAvg. How was this value tuned?
- FedAvg + FT is not a strong baseline for DP. For a more compelling case, it would be good to see at least 3 more baselines: (a) individual training per client without communication, (b) Consensus-based full personalization (e.g. Ditto) with DP, and (c) Shared representation learning with DP. Similarly, for FedAvg + FT, what happens if one tunes the number of personalization rounds?

**Clarity**: Some minor comments that might help improve clarity:
- The 2nd half of page 2 might read much smoother if the regularization term $H_i(u_i, u_0)$ is described.
- Why feature the partial variance reduction so prominently here in the intro if it is not discussed at length in the work? A suggestion (which the authors can feel free to ignore) is to phrase this as an experimental ablation with the corresponding theory in the appendix.
- A short half-sentence about each experimental baseline would be helpful in Section 6.1.

**Questions:**

1. Do the authors wish to highlight novel technical challenges they have had to overcome in their proofs?
2. Do the experiments use partial or full client participation?
3. The communication size of FedReCo is highlighted in several cases. However, I would guess that the representation learning part of the network constitutes the vast majority of the paper of the parameters, so the improvements in terms of communication are marginal at best (for the experimental settings considered). How much precisely does one save on communication by not sending the last layer?
4. All the above questions about the DP experiments.

---

### Official Review · Reviewer_R9Ht · 2023-11-10

**Soundness:** 3 good
**Presentation:** 3 good
**Contribution:** 3 good
**Rating:** 5
**Confidence:** 3

**Summary:**

The paper combines the consensus optimization formulation of federated learning with the idea of (separation of) global feature extraction and local prediction to come up with a consensus regularized personalized federated learning at representation level.

**Strengths:**

The paper presents both theoretical and experimental support for the method proposed.

The method is well presented, readers can understand easily the idea behind FedReCo.

**Weaknesses:**

Not sure what is the benefit of proposed FedReCo over Ditto. Looks like from Table 2 and Figure 3(a), Ditto is fast and accurate.

The experimental part is not comprehensive enough. For example, not sure how many random seeds have been run for each method. And I only see a simple figure of accuracy on CIFAR10, feels like a lack of more experiments (maybe on more datasets that is common in FL, e.g. the LEAF benchmarks).

**Questions:**

The differential privacy part is a bit abrupt and immature, not enough details or descriptions regarding that part. Maybe it is better to not include it.

I am not fully convinced that combining the two ideas (consensus regularization and representation network) is a superior way to do personalized federated learning, it might even feel redundant since they are different approaches to personalized FL.

---

### Author Response · Authors · 2023-11-20

We thank all of the reviewers for their insightful and constructive comments. We have decided to withdraw the paper because we found that there is a special case in the experiments where local training itself can obtain good performance, meaning that there is no need to collaborate with other clients. Thus we need to modify our proposed algorithm and evaluate its performance on more settings and datasets (reason below). Nonetheless, we chose to answer some key questions from the reviewers, and will implement their suggestions in our future study.

$\textbf{Convergence Rate}$: In our algorithm, only $u_0$ is trained at server from all the clients' local information. Whereas $u_i$ and $v_i$ are trained only locally, thus not enjoying the benefit of linear speedup in distributed learning. We can see the training of $u_0$ has the linear speedup in theoretical result, but for the general result for all the three terms, there is no linear speedup and only $O(\frac{1}{\sqrt{T}})$ rate.

$\textbf{Partial Participation}$: The theoretical analysis and experimental setting can be easily extended to partial participation situation without any additional cost. In the future version we will consider adding the content with partial participation, though putting these extra constraints may make the main message obscure.

$\textbf{Regularization Term}$: We found the main problem in our algorithm to be that $u_0$ is purely trained by the regularization term $H_i(u_0, u_i)$. It seems now that the regularization term is too weak to train the $u_0$ from local information in each client. Actually $u_0$ cannot effectively learn local knowledge from the regularization term only. We have found a way to fix this to get better performance, that we will incorporate in future versions.